# Slightly beneficial genes are retained by bacteria evolving DNA uptake despite selfish elements

Bram van Dijk[1†]*, Paulien Hogeweg[1]*, Hilje M Doekes[1], Nobuto Takeuchi[2]

[1]Utrecht University, Theoretical Biology, Utrecht, Netherlands; [2]University of Auckland, Biological Sciences, Auckland, New Zealand

**Abstract** Horizontal gene transfer (HGT) and gene loss result in rapid changes in the gene content of bacteria. While HGT aids bacteria to adapt to new environments, it also carries risks such as selfish genetic elements (SGEs). Here, we use modelling to study how HGT of slightly beneficial genes impacts growth rates of bacterial populations, and if bacterial collectives can evolve to take up DNA despite selfish elements. We find four classes of slightly beneficial genes: indispensable, enrichable, rescuable, and unrescuable genes. Rescuable genes — genes with small fitness benefits that are lost from the population without HGT — can be collectively retained by a community that engages in costly HGT. While this 'gene-sharing' cannot evolve in well-mixed cultures, it does evolve in a spatial population like a biofilm. Despite enabling infection by harmful SGEs, the uptake of foreign DNA is evolutionarily maintained by the hosts, explaining the coexistence of bacteria and SGEs.

**\*For correspondence:**
bramvandijk88@gmail.com (BD);
p.hogeweg@uu.nl (PH)

**Present address:** †Theoretical Biology, Utrecht University, Utrecht, Netherlands

**Competing interests:** The authors declare that no competing interests exist.

## Introduction

Horizontal Gene Transfer (HGT), the transmission of genetic material between unrelated individuals, is a major factor driving prokaryotic evolution (*Ochman et al., 2000*; *Doolittle and Zhaxybayeva, 2009*; *Vogan and Higgs, 2011*). Recent estimates of the rate of HGT in closely related bacteria are staggeringly high (*Iranzo et al., 2019*; *Sakoparnig, 2019*), with HGT possibly even outpacing gradual sequence evolution (*Hao and Golding, 2006*; *Puigbò et al., 2014*; *Vos et al., 2015*). Combining this with the fact that prokaryotes adapt mostly through rapid gene loss (*Kuo and Ochman, 2009*; *Morris et al., 2012*), bacterial adaptation appears to be mainly driven by changes in gene content (*Snel et al., 2002*; *Treangen and Rocha, 2011*; *Nowell et al., 2014*). Rather than waiting for rare beneficial mutations to arise, taking up tried-and-true genes from a shared 'mobile gene pool' allows bacteria to adapt quickly to different ecological opportunities (*Jain et al., 2003*; *Wiedenbeck and Cohan, 2011*; *Casacuberta and González, 2013*; *Mell and Redfield, 2014*; *Niehus et al., 2015*; *Lopatkin et al., 2016*). Indeed, many bacterial species show patterns consistent with this rapid turnover of genes, where strains from a single niche contain a relatively small set of genes, while the set of genes found by sampling strains from various niches (i.e. the pan-genome) is much richer (*Welch et al., 2002*; *Lefébure and Stanhope, 2007*; *Touchon et al., 2009*; *Kim et al., 2015*). Hence, genes appear to be rapidly lost from any individual lineage, but are retained in a much larger gene pool through HGT.

When considering the effects of HGT on gene content, it is important to note that HGT can add novel genes to the genome. This might happen through plasmid transfer (i.e. conjugation), but also through recombination. For example, when bacteria take up DNA from their environment (i.e. transformation), genes inserted between two homologous regions may be integrated into the genome. The DNA that has been taken up may also carry mobile genetic elements (MGEs), which can integrate into the genome without a requirement for sequence similarity. Such processes have been

**eLife digest** Most animals, including humans, inherit genes from their parents. However, bacteria and other microorganisms can also acquire genes from members of the same generation. This process, called horizontal gene transfer (HGT for short), allows bacteria to quickly adapt to new environments. For example, rather than waiting for rare mutations to arise, bacteria can pick up 'tried and true' genes from their neighbours which allow them to exploit new resources or become resistant to antibiotics.

But gene sharing comes at a cost. For instance, taking up DNA is an energetically costly process and exposes bacteria to so-called selfish genes which replicate at the expense of other more useful genes in the genome. Given the costs and the threat of selfish genes, it remained unclear whether HGT is still beneficial in a stable environment where no new resources or antibiotics are present.

Here, van Dijk et al. used mathematical modelling to examine how gene sharing affects the growth rate of bacterial colonies living in a stable environment. The experiments showed that bacteria are able to take up new sequences of DNA even in the presence of selfish genes. This allows communities of bacteria to retain genes that provide a small benefit that would otherwise be lost from the population, even when taking up DNA imposes a cost upon the individual. van Dijk et al. found that this collective behaviour cannot evolve in well-mixed bacterial populations, but readily emerged in more structured populations, such as biofilms.

This work demonstrates how HGT, a key component of bacterial evolution, has allowed bacteria to coexist with harmful selfish genes. It also provides insights into how genes persist and spread through bacterial communities, which has implications for our understanding of antibiotic resistance.

coined 'additive HGT' (*Thomas and Nielsen, 2005*; *Choi et al., 2012*; *Soucy et al., 2015*), which is distinct from 'replacing HGT' because of its ability to *copy* genes from one individual to another. On the one hand, DNA uptake may be beneficial for the cells, as it allows adaptations to new environments (*Casacuberta and González, 2013*; *Mell and Redfield, 2014*; *Lopatkin et al., 2016*) or the recovery of lost genes (*Vogan and Higgs, 2011*; *Takeuchi et al., 2014*). On the other hand, DNA uptake also poses a risk in the form of Selfish Genetic Elements (SGEs) such as transposons, and may furthermore cause chromosome disruptions and cytotoxicity (*Baltrus, 2013*). Finally, the translocation of DNA and the required state of competence generally comes at energetic and metabolic costs for the cell in terms of expressing and operating DNA uptake machinery (*Ephrussi-Taylor and Freed, 1964*; *Bergé et al., 2017*; *Villa et al., 2019*). Hence, while picking up genes can be very beneficial for bacteria when adapting to a new environment, taking up foreign DNA is also a costly and highly risky endeavour (*Vogan and Higgs, 2011*; *Baltrus, 2013*). Given these disadvantages, is the uptake of DNA ever adaptive for bacteria when the environment does *not* change? Can HGT be considered an evolved trait of bacteria, or is it only a side-effect of other unrelated processes like infection by SGEs or DNA repair (*Redfield, 2001*)?

To address these questions, we here present and analyse a model of a bacterial population undergoing HGT of a single gene. We consider HGT through the uptake of genes from a shared pool of DNA, that is bacterial transformation. We assume that HGT is a costly process for the host cells, and that these costs are proportional to the rate of uptake. Such costs may reflect energetic costs for translocating DNA, growth impediments during the state of natural competence, or the various risks of incorporating foreign DNA into the genome. We show that this form of HGT has a positive impact on population growth rates by recovering slightly beneficial genes, which are hard to maintain in the population through selection alone. Based on whether or not the genes are lost from the population without HGT, and whether HGT can improve the population growth rate, we find that genes fall into one of five gene classes: (i) *indispensable genes*, that are never lost from the population, and for which HGT is therefore unnecessary and deleterious, (ii) *enrichable genes*, that are not lost from the population, but enriching the genes via HGT can nevertheless improve growth rates, (iii) *rescuable genes*, which are lost from the population without HGT, but can be rescued by HGT which improves population growth rates, (iv) *unrescuable genes* which are also lost from the population without HGT, but recovering them with HGT does not improve growth rates, and (v) *selfish genetic elements*, which confer a fitness penalty but can persist through HGT. For enrichable and

rescuable genes, where HGT can increase population growth rates, we also investigate if HGT, that is the ability of cells to take up DNA, can evolve de novo. While the bacteria readily evolve to use HGT for enrichable genes, having sufficient donor cells to interact with, evolving HGT to 'rescue' rescuable genes faces a problem: HGT is needed for the gene to persist in the population, but sufficient donor cells are required to make HGT adaptive. This paradox is however resolved in a spatially structured population like a biofilm, as even a minority of donor cells can be locally abundant, giving rise to a localised 'gene-sharing' community that eventually overgrows the whole population. Finally, in this spatial eco-evolutionary context, HGT is evolutionarily maintained even when exploited by harmful genetic parasites, resulting in stable coexistence of bacteria and SGEs. Our model provides important insights and search images for how slightly beneficial genes may spread, or fail to spread, in an evolving microbial population.

## Results

Throughout this study, we analyse how HGT affects the growth rates of bacterial populations, and to what extent bacteria can evolve to engage in HGT by evolving DNA uptake. We consider a 'hard case', where HGT is a continuously costly process for individual cells and is only beneficial under specific circumstances. Consider two cell types: cells that carry a beneficial gene (carriers, C), and cells that do not (non-carriers, N). The benefit of carrying the gene, $b$, makes carriers grow faster than non-carriers (or slower if $b < 0$, e.g. the gene is a selfish element), but carriers lose the beneficial gene at a fixed rate $l$. Non-carriers can recover genes by interacting with carriers through HGT. We study these dynamics with different models, first using simple ordinary differential equations (ODEs, *Figure 1A,B*) and later an individual-based model (IBM) that takes spatial population structuring into account (*Figure 1C*). The equations and full description of the models can be found in the Methods section.

Starting with the simplest model depicted in *Figure 1A*, we first illustrate how the steady-state frequency of carrier cells depends on the benefit of the gene ($b$) and the rate of HGT ($h$). *Figure 2A* shows that if the gene is sufficiently beneficial, most of the population will consist of carrier cells with or without HGT. Despite being continuously lost, these genes are beneficial enough to readily persist in the population through selection. An increased rate of HGT results in only marginally more carrier cells. For genes with a much smaller benefit, HGT can have a large impact on the frequency of carrier cells in the population. In fact, if the benefit is very small ($b<l$, white dotted line), carriers do not survive in the absence of HGT at all, but can occur in fairly high frequencies with sufficient HGT. Note however that the mere survival of carriers with beneficial genes does not imply a positive impact on the population growth rate, as the model assumes HGT comes at a cost. Actually, at sufficiently high rates of HGT, carrier cells with costly genes ($b<0$) can also persist in the population, which by definition is deleterious for growth. These costly genes could either be genes that are expressed but not useful in the current environment, or Selfish Genetic Elements (SGEs). Throughout this study, we consider genes with $b<0$ to be SGEs.

### Slightly beneficial genes fall into distinct gene classes

To better understand the impact of HGT, we next study how HGT impacts the population growth rate ($\phi$). The population growth rate in steady state is given by *Equation 1* displayed below (see full derivation in Supplementary Section 1). The function is comprised of two parts; one where the population consists only of non-carriers (if $h \leq l - b$), and one where carriers survive and the gene persists within the population (if $h > l - b$). When the gene persists, an optimal growth rate is found at $h_{opt} = \sqrt{bl/c} - b$ (see Supplementary material).

$$\phi^*(h) = \begin{cases} 1 - ch & \text{if } h \leq (l-b)\,(\text{gene cannot persist}) \\ 1 - ch + b - \frac{bl}{b+h} & \text{if } h > (l-b)\,(\text{gene persists}). \end{cases} \tag{1}$$

By analysing *Equation 1*, we find that we can distinguish distinct classes of genes depending on (i) whether HGT is required for the gene to persist within the population, and (ii) whether HGT is beneficial for the population growth rate (*Figure 2B*). When genes are highly beneficial ($b > l/c$), HGT is not required for the gene to persist, and HGT does not improve the population growth rate. In other words, although transferring these *indispensable genes* yields a small increase in the

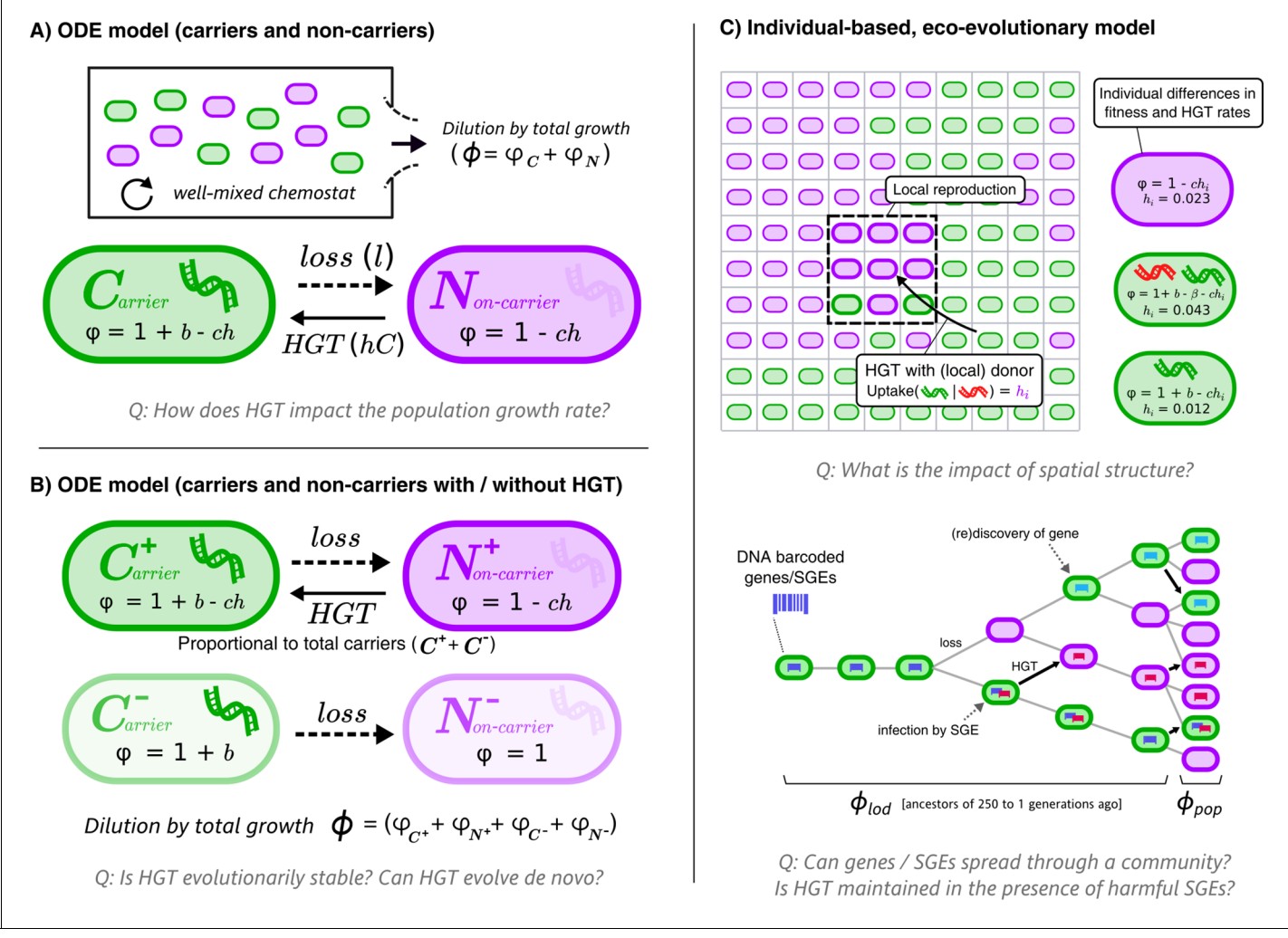

**Figure 1.** Graphical overviews of the different models: This study uses a series of models with gradually increasing complexity. The first two models (**A**+**B**) are composed of Ordinary Differential Equations (ODEs), and the third model (**C**) is an individual-based model (IBM). The models describe a population of bacterial cells which either carry a beneficial gene (carriers, *C*) or do not carry the genes (non-carriers, *N*), both of which engage in costly HGT (e.g. take up DNA or enter a state of competence). The cells are competing for a limited resource, where the intrinsic growth is 1, *b* is the growth rate advantage (or disadvantage) for carrying the gene, *l* is the rate at which the gene is lost, *h* is the rate of HGT (*i.e.* uptake/competence), *c* is the cost of HGT, $\varphi$ is the growth rate of sub-populations/individual cells, and $\phi$ represents the total growth rate. In the IBM, each cell has an individual rate of HGT (arbitrary values are shown in the cartoon), which we use to study the de novo evolution of HGT. The IBM also makes a distinction between the average growth rate of the population ($\phi_{pop}$) and the average growth rate of the line of descent ($\phi_{lod}$, previous 250 generations of cells). In the IBM, both beneficial genes (with benefit *b*, green) and harmful selfish genetic elements (SGEs, red) with a fitness penalty $\beta$ are taken into account. Genes and SGEs are tagged with a unique barcode when they flux in, which are inherited upon reproduction or transfer. Parameters *c*, *h* and *l* are assumed to be positive.

number of carrier cells, this does not outweigh the costs of HGT. When considering lower values of *b*, HGT is still not required for the gene to persist within the population, but transferring these *enrichable genes* is nevertheless beneficial for population growth rates. For even lower benefit (*b* < *l*), HGT is a necessity for the gene to persist within the population, but the population growth rate can be improved by means of intermediate rates of HGT. We call these genes *rescuable genes*. If we consider genes with even smaller fitness effects ($b < 4cl/(1+c)^2$), HGT is still required for the survival of these genes, but the population growth rates are highest in the absence of HGT. Thus, despite being defined as a beneficial gene (*b* > 0), transferring these *unrescuable genes* is not beneficial. Finally, we can consider SGEs, genes with a negative effect on fitness (*b* < 0). These genetic parasites can only persist in the population at very high rates of HGT, but are of course never

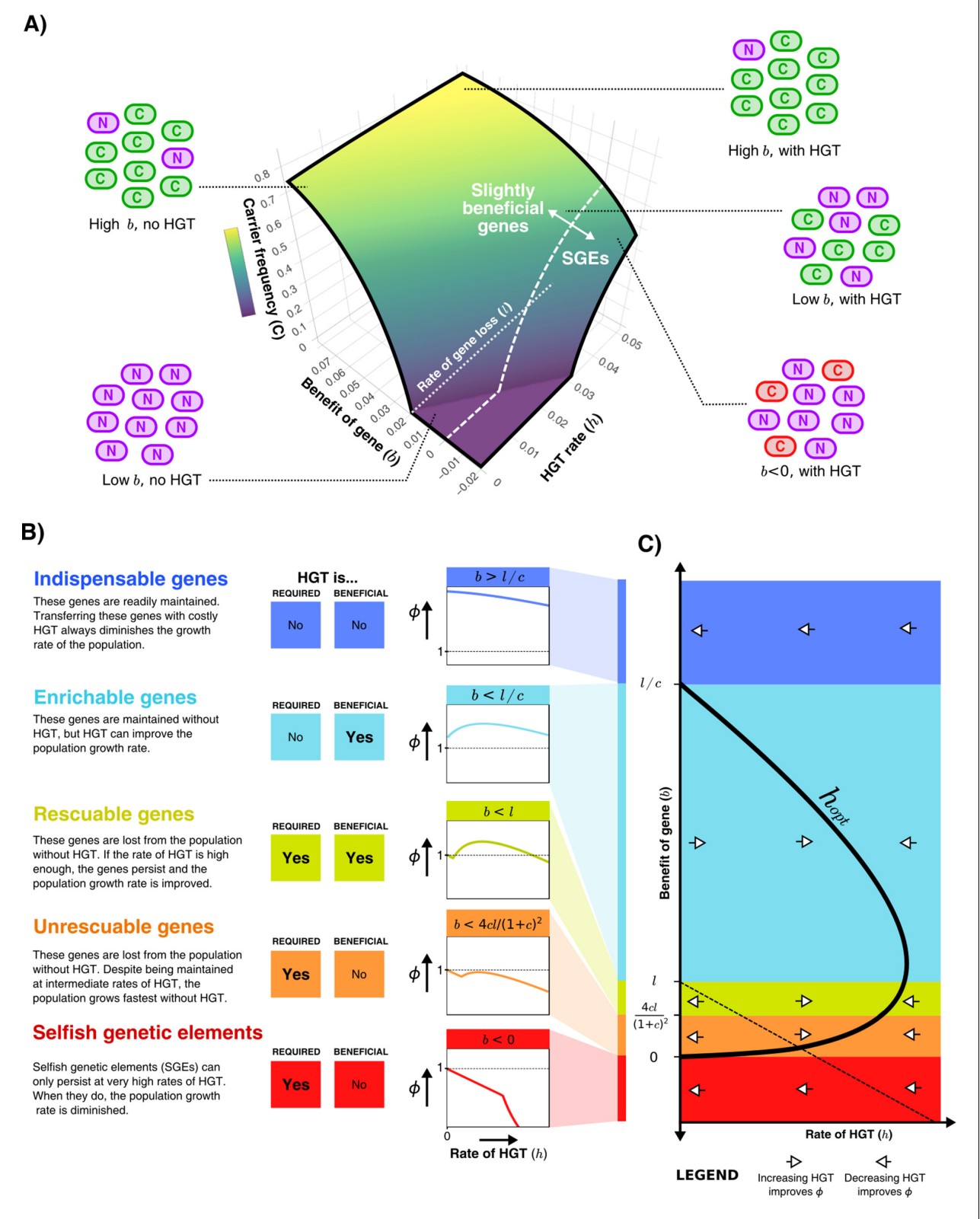

**Figure 2.** HGT can help genes persist in the population, resulting in distinct gene classes of slightly beneficial genes. (**A**) For the model shown in *Figure 1A*, the frequency of carrier cells is shown in a 3D surface plot for different values of $b$ and $h$. This function is derived in the Supplementary Material, and here drawn for $l = 0.02$ and $c = 0.2$. The white dashed line ($b = 0$) gives the boundary between slightly beneficial genes and selfish genetic elements (SGEs). Cartoons illustrate how, for a very beneficial gene (high $b$), HGT leads only to a mild increase in carrier cells, how HGT has a large

Figure 2 continued

impact when the gene brings a smaller fitness effect (low $b$), and how SGEs can also persist with high HGT rates ($b < 0$). (B) Different classes of slightly beneficial genes can be distinguished based on (i) if HGT is *required* for the gene to persist within the population (when $b < l$) and (ii) if HGT is *beneficial* for population growth rates. The graphs on the right-hand side show, for each of these classes, how an increasing rate of HGT (x-axis) influences the population growth rate $\phi$ (y-axis). (C) A bifurcation diagram shows how the population growth rate is either improved or diminished by HGT for different values of the rate of HGT ($h$, x-axis) and the benefit parameter ($b$, y-axis). The HGT rate that optimises population growth rates ($h_{opt} = \sqrt{bl/c} - b$) is depicted by the thick black curve. The dashed line is given by $h = l - b$, above which the genes are able to persist in the population. Finally, white arrows depict whether $\partial\phi/\partial h$ is positive or negative, indicating how more/less HGT changes the population growth rate.

beneficial for the population growth rate. *Figure 2C* shows a bifurcation diagram that summarises how increasing or decreasing rates of HGT impact the population growth rate for these different classes.

## HGT is an evolutionarily stable strategy, but cannot evolve to 'rescue' rescuable genes

By analysing the simple model of cells undergoing HGT, we have found five distinct gene classes. For two of these classes, namely enrichable and rescuable genes, moderate rates of HGT improve the population growth rates. We next study (i) whether HGT of enrichable and rescuable genes is an evolutionarily stable strategy, and (ii) if bacteria can evolve this strategy de novo. To answer these questions, we consider two competing species: one that takes up DNA, and one that does not ($HGT^+$ and $HGT^-$ respectively, see *Figure 1B*). With this model, we have studied the evolution of DNA uptake by means of adaptive dynamics (*Metz et al., 1995*). If $HGT^-$ cannot invade $HGT^+$, we call HGT an evolutionarily stable strategy, and if $HGT^+$ can invade $HGT^-$ we call HGT evolvable.

We found that HGT is an evolutionarily stable strategy for both enrichable and rescuable genes, but that HGT is evolvable only for enrichable genes (see Supplementary Material for full analysis). In other words, invasion of $HGT^+$-mutants is not possible with respect to rescuable genes. Even when we assume that the invading $HGT^+$-mutant has the optimal rate of HGT, it cannot invade into a population of $HGT^-$ cells in steady state. These results were confirmed by numerical analysis, which indeed shows that $HGT^+$ only invades when the founding population size of $HGT^+$ ($C^+/N^+$) is relatively large (see *Figure 3A*). This failure to reach the alternative (fitter) evolutionary attractor is caused by positive frequency-dependent selection (known as the Allee effect). Invading mutants, that is a small population of $HGT^+$ cells, contain few carrier cells to act as donors for HGT. Moreover, since the resident population of $HGT^-$ is also not able to retain the rescuable genes, the resident population can also not serve as a donor (see *Figure 3B*). As such, the costs of HGT for an invading $HGT^+$-mutant do not outweigh the potential benefits. In summary, while HGT is an evolutionarily stable strategy, cells cannot evolve HGT to 'rescue' rescuable genes.

## Spatial structure hinders the maintenance of genes, making HGT adaptive for a wider range of genes

So far, we have studied a well-mixed population of cells that undergoes all-against-all competition, and found that HGT is advantageous for slightly beneficial genes that (i) are not too beneficial, as these genes readily persist within the population without HGT, and (ii) are beneficial enough to compensate for the costly HGT. Next, we study the same dynamics of carrier and non-carrier cells in a spatially explicit, eco-evolutionary context. We do this by implementing an individual-based model (IBM), where bacterial cells reside on a grid, interactions are local, and events like HGT and gene loss are implemented as stochastic processes (see Materials and methods and *Figure 1C*). When the cells on this grid are sufficiently mixed each time step, the IBM should approximate the dynamics of the ODE model. However, when cellular mixing is minimal, the resulting spatially structured population is more analogous to that of a biofilm. What is the effect of this spatial structure?

We first analysed the IBM for a wide variety of values for $b$ and $h$, and measured the average growth rates $\phi$ in the population. We can thus evaluate whether the aforementioned gene classes (indispensable, enrichable, rescuable, unrescuable genes, and SGEs) are found under the same conditions as in the ODE model. *Figure 4A* shows that, when the IBM is well-mixed, the gene classes indeed occur at values of $b$ identical to the ODE model. However, the gene classes shift to higher values of b when mixing is decreased, making the range of benefits in which genes are classified as

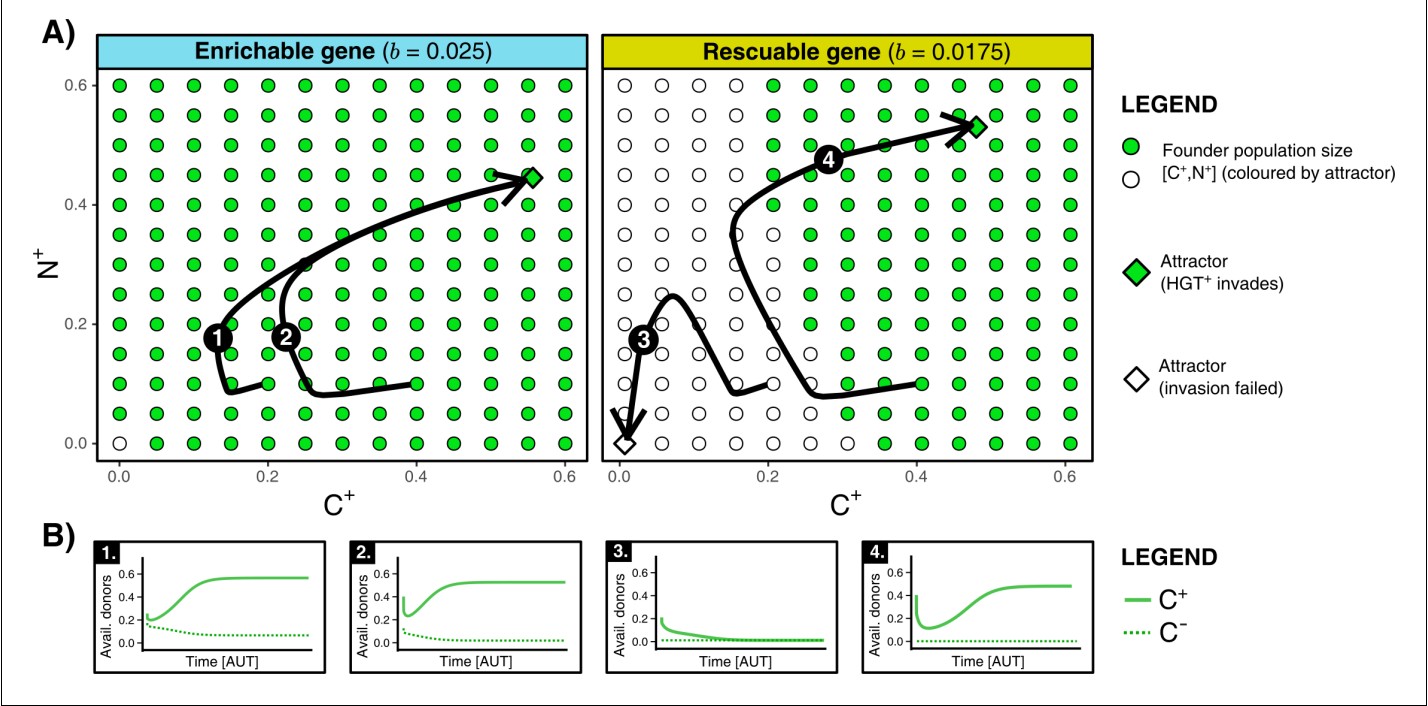

**Figure 3.** HGT is an evolutionarily stable strategy but is evolutionarily inaccessible for rescuable genes due to a lack of gene-carrying donor cells. **(A)** For an enrichable and a rescuable gene ($b = 0.025$ and $b = 0.0175$ respectively), a 2D projection of the 4D state space is shown. For various founder sizes (combinations of carriers, $C^+$, and non-carriers, $N^+$), the result of invasion of $HGT^+$ (that has the optimal rate of HGT, $h = h_{opt}$) into $HGT^-$ ($h = 0$) is shown. $C^+/N^+$-combination that successfully invade are annotated as green dots, while failed invasions are coloured white. $HGT^+$ always successfully invades for enrichable genes. For a rescuable gene, low founder populations sizes of $C^+$ and $N^+$ (white dots) fail to invade, whereas they can invade at higher population sizes (green dots). Black arrows (1-4) show the trajectories starting from two founder population sizes. **(B)** For the four trajectories from panel A, the graphs show the temporal dynamics of gene-carrying donor cells.

enrichable and rescuable much broader. In these biofilm populations, HGT was indeed found to be evolutionarily stable for this wider range of fitness-effects (black outline), illustrating that it is not only the value of $b$, but also the ecological context in which a gene finds itself that determines whether or not HGT is adaptive.

What causes these gene classes to shift depending on this spatial context? How does an enrichable gene in the well-mixed system become rescuable in the spatially structured population, as though it is less beneficial? *Figure 4B* shows how this can be intuitively understood by taking into account how, with low mixing, individuals in a spatial system mostly compete with their own kind (i.e. progeny and conspecifics). Even when the majority of the population consists of non-carriers, carriers are still competing mostly with other carrier cells. Thus, the effective benefit of carrying the gene is lower in a biofilm, hence the gene becomes harder to maintain within the population. In *Figure 4C* is shown that, while carrier cells in well-mixed population experience a competitive advantage of 2% when carriers make up approximately half the population, carriers in a biofilm only reach a similar competitive advantage at very low carrier frequencies, that is when the carriers are almost extinct. At this point, the gene will readily be lost stochastically. The hampered ability of spatially structured populations to retain slightly beneficial genes, indeed changes how the population growth rate depends on the rate of HGT (*Figure 4D*).

## Bacteria evolve DNA uptake for rescuable genes only in a spatially structured population

The results described in the previous section illustrate that HGT (i.e. the uptake of DNA) is an evolutionarily stable strategy for a much broader range of $b$-values (fitness effects of genes) in a spatially structured population than in a well-mixed culture. Many more genes are furthermore classified as

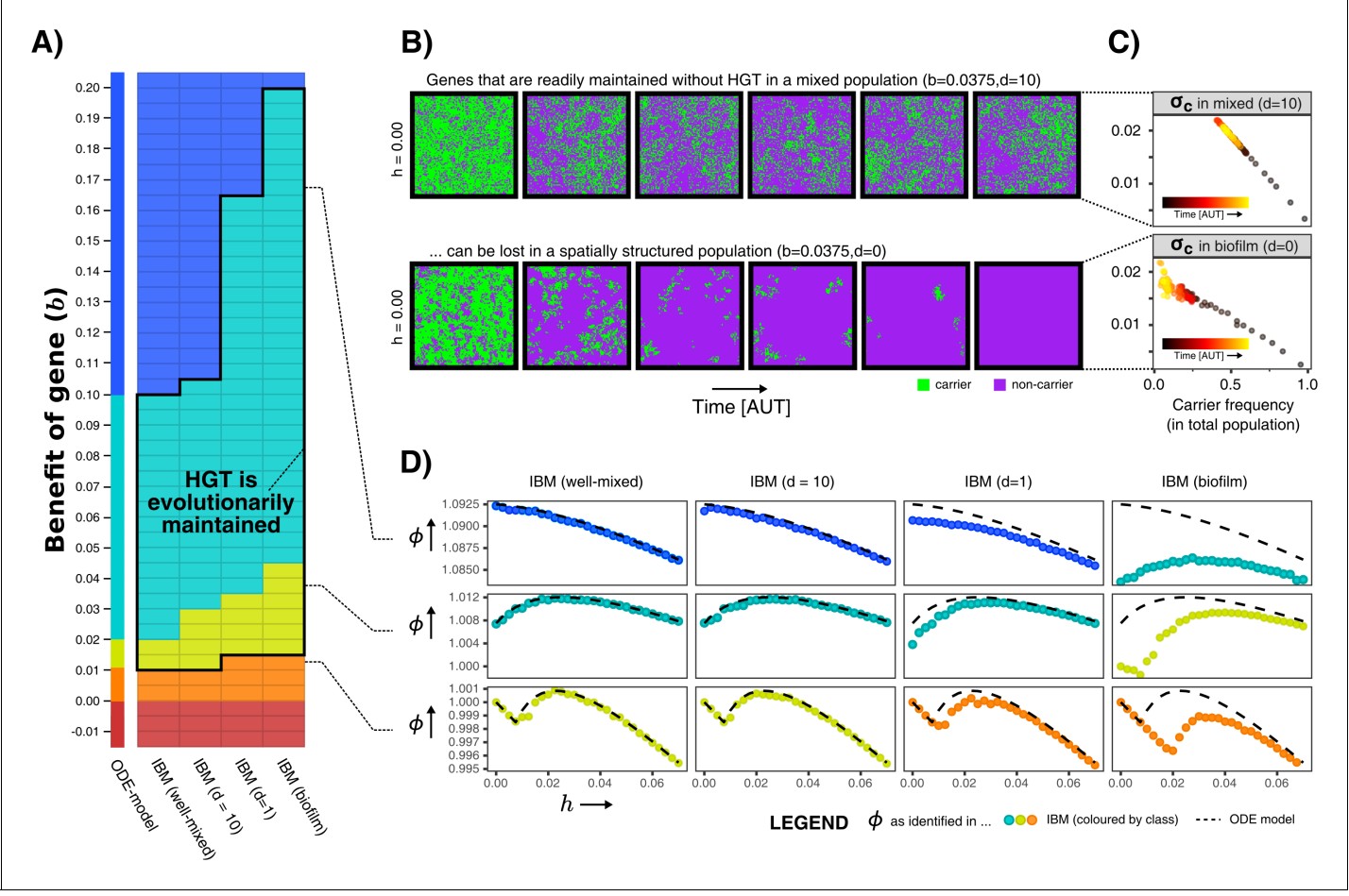

**Figure 4.** Spatial structure hinders the maintenance of slightly beneficial genes. (**A**) Each tile in this table represents a series of simulations in the individual-based model (IBM), where we first test which gene class (background colour) is found when sweeping over different HGT-rates ($h$-values), and next test whether HGT (*i.e.* DNA uptake) is evolutionarily maintained when starting with a population consisting of only carrier cells with $h = 0.05$ (shown with black outline, see *Appendix 1—table 1*). This was tested for the well-mixed IBM and the IBM with different levels of mixing ($d$). The continuum of gene classes from the ODE-model is presented for comparison. Colours are the same as in *Figure 2* (blue = indispensable, cyan = enrichable, yellow = rescuable, orange = unrescuable, red = selfish genetic element). (**B**) Shown is the spatial grid of the IBM for two simulations with the same value of $b$, and no HGT. The gene readily persists in the mixed IBM (top panel, $d = 10$), while the gene does not persist in the spatially structured population (bottom panel, $d = 0$). (**C**) For the simulations shown in B, we plot $\sigma_c$ (see Materials and methods) against the frequency of carrier cells. This value of $\sigma_c$ indicates how much fitter carrier cells are than their local competitors, which decreases to 0 when all local competitors are also carriers. When comparing the top and bottom panel, this shows how clumping hinders the effective benefit of carrying a gene. (**D**) For three rows from the table of A, it is illustrated how the effect of spatial clumping illustrated in B and C modifies the gene class found for specific $b$-values. The dashed line indicates the growth rates predicted by the ODE model.

rescuable in these spatially structured populations, meaning that they can only persist through HGT. We have concluded in a previous section that HGT cannot evolve to 'rescue' these rescuable genes in populations that are well-mixed, fully deterministic, and by only considering a single $HGT^+$ mutant type at a time. In the IBM on the other hand, the population is spatially structured, events are stochastic, and each individual cell has its own rate of DNA uptake. Can these different assumptions help to alleviate the Allee effect mediated by a lack of donor cells, which prevents the de novo evolution of HGT?

To answer the question posed above, we allowed the HGT-rate $h$ (i.e. the costly uptake of DNA) of all individuals in the IBM to evolve (see Materials and methods). When a non-carrier interacts with a (local) carrier, the $h$-value of this non-carrier (i.e. the recipient) determines the probability of accepting the gene. For simplicity, we will call individuals with an $h$-parameter greater than 0.02 $HGT^+$, and the others $HGT^-$. We start with a non-carrier population of $HGT^-$ cells (with $h = 0.00$),

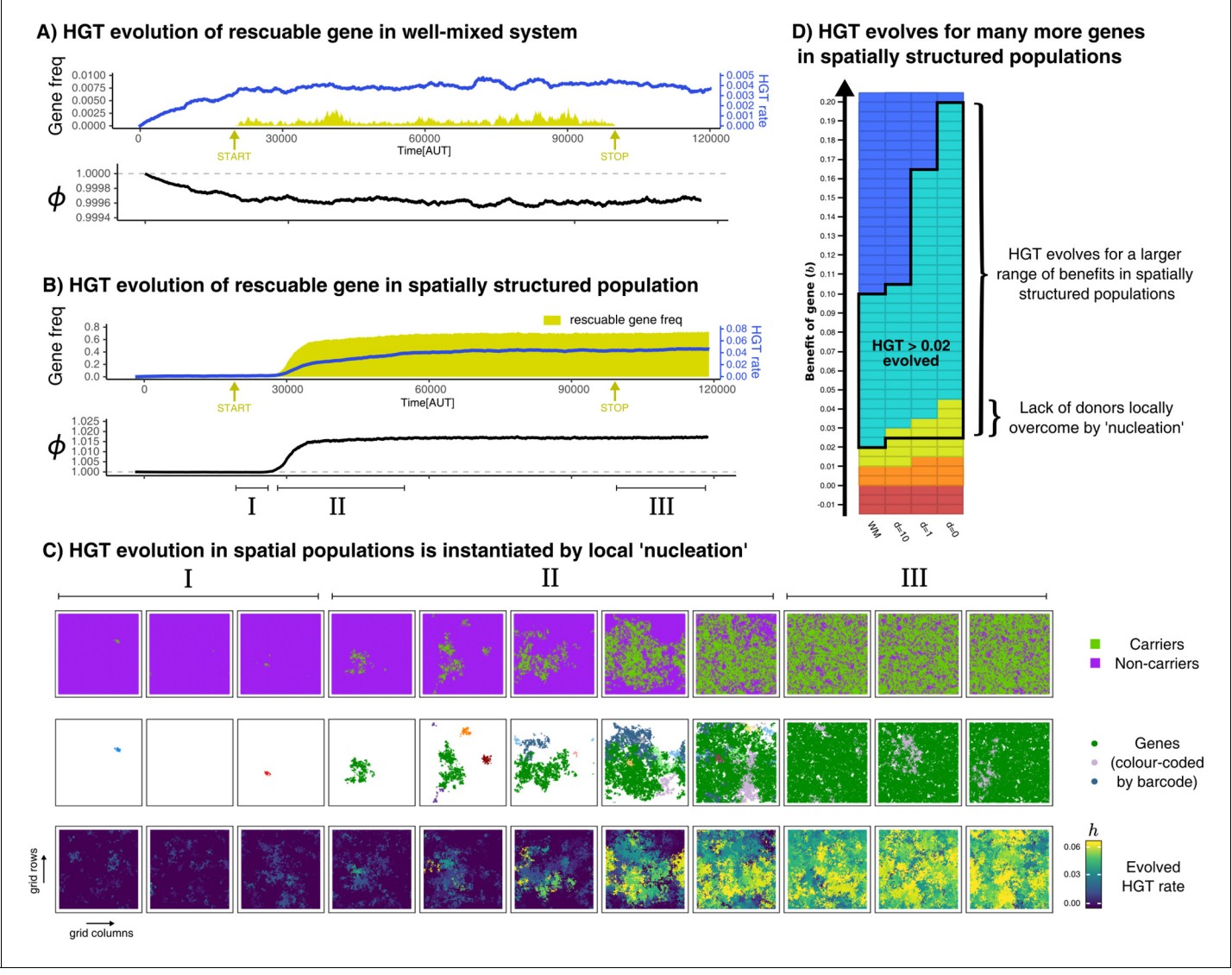

**Figure 5.** HGT of rescuable genes only evolves in spatially structured populations due to the emergence of 'gene-sharing' communities. (A-B) Both panels show the frequency of a rescuable gene (yellow area) that is discovered with a very low probability ($5 \cdot 10^{-6}$ per time step), the mean evolved rate of DNA uptake within the population (blue line), and the growth rate of the population (black). Note that A and B have a different range in the y-axis for clarity. (C) Different visualisations of the grid show, in the spatially structured populations, carrier cells with a rescuable gene (colour coded by the unique barcodes) spread after a local 'nucleation event'. A positive feedback loop follows, resulting in a 'gene-sharing' community which slowly overgrows the rest of the population. (D) The outcome of de novo HGT evolution for the same combinations of fitness-effects and mixing as in *Figure 4A*. Parameters used: $h_{init} = 0.0$, $u = 5 \cdot 10^3$, $m = 0.05$, $l = 0.02$, $c = 0.1$, $f = 5 \cdot 10^{-6}$, $f_{start} = 20,000$, $f_{stop} = 100,000$, $n = 400$ (*i.e. $N = 400^2$*). For the well-mixed population, we chose a rescuable gene with the highest benefit from *Figure 4A* ($b = 0.0175$), and for the spatially structured population we used $b = 0.030$ (the average of the much broader range of rescuable genes). Both these genes require HGT to persist, and are thus rescuable within their own spatial context.

simulate this population for some time (20,000 time steps), and then allow cells to sporadically discover rescuable genes. Since rescuable genes cannot persist without HGT, the fate of a recently discovered gene depends on the ability of cells to engage in (local) HGT. Using this protocol, we investigated if the rescuable gene is able to spread through the evolution of HGT. We found that HGT never evolved for rescuable genes in well-mixed populations (*Figure 5A*), consistent with our prior results in the well-mixed ODE model. Thus, we can conclude that the level of stochasticity in the IBM is insufficient to overcome the aforementioned Allee effect caused by a lack of donor cells.

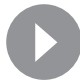

**Video 1.** Gene-sharing 'nucleation events' and long-term coexistence with harmful SGEs. For the IBM-model, this Video shows how gene-sharing of rescuable genes emerges through a 'nucleation'-event, allowing local communities to eventually overgrow all other cells. The top-left panel shows carriers/non-carriers, the top-right shows the evolved HGT-rate (i.e. DNA uptake), the bottom-left shows the barcodes of influxed genes, and the bottom-right shows the barcodes of influxed SGEs.

https://elifesciences.org/articles/56801#video1

In the spatially structured population, HGT of rescuable genes *does* in fact emerge, therewith 'rescuing' the rescuable genes (*Figure 5B*). Interestingly however, we found that HGT did not always evolve immediately after the influx of rescuable genes started (yellow arrow), but nevertheless spread steadily once attained. To further elucidate the spread of genes, we barcoded each newly discovered gene with a unique ID, and visualised these on the spatial grid with different colours (*Figure 5C*). Initially, rescuable genes fail to invade, even though different barcodes may locally persist for a while (episode I). After some time, however, one gene (green) manages to persist within a local community of transferring cells (episode II). This sets in motion a positive feedback mechanism, where the local abundance of the green gene alleviates the lack of donor cells, transforming nearby $HGT^+$-mutants into carriers, and so on (also see *Video 1*). This emergent 'gene-sharing' community eventually overgrows the other cells, and the rescuable gene ultimately persists in up to 70% of the population. After the influx of rescuable gene is stopped (episode III), the gene readily persists within the population, showing how this transferring community does not depend on the continuous influx of genes. In summary, HGT of rescuable genes through DNA uptake can only evolve if transfer happens within spatially localised sub-populations, and not under well-mixed conditions modelled by mass-action. Through a local 'nucleation event', communities can reach the alternative stable state that can maintain the rescuable gene. *Figure 5D* summarises the outcome of de novo HGT evolution for a broad range of genes (*b*-values) with different levels of mixing, revealing how the uptake of DNA evolves for many more genes in a spatially structured population. While HGT of enrichable genes always evolved, HGT only evolved for rescuable genes in spatially structured populations. Finally, as expected from prior results, HGT never evolved for indispensable and unrescuable genes.

## HGT is evolutionarily maintained in the presence of harmful SGEs

We have shown that HGT can be adaptive and evolvable for bacteria in order to enrich or rescue slightly beneficial genes. We next investigated if HGT can be maintained under the pressure of harmful SGEs, genetic parasites that spread through horizontal transfer. Our earlier analysis has shown that, when the rate of HGT cannot evolve, SGEs (genes with $b<0$) can persist within the population as long as $h>l-b$. However, when the rate of HGT is allowed to evolve, bacteria may lower their rate of DNA uptake to avoid these genetic parasites. Therefore, we next investigate whether bacteria in the IBM will maintain their ability to take up foreign DNA in the presence of SGEs. For this, we consider a population that evolved to engage in HGT of a rescuable gene ($b$=0.03), and expose this population to a low influx of SGEs which confer a fitness penalty ($\beta$). We study if these SGEs, despite their fitness penalty, can persist within this bacterial population, and if HGT is evolutionarily maintained by the hosts. *Figure 6A* shows that, when the fitness penalty of the SGEs is small relative to the benefit of the rescuable gene (hereafter called 'weak SGEs', $\beta = 0.01$), these genetic parasites quickly rise to very high frequencies within the population. Although the host cells gradually evolve lower HGT rates in response (from $h \approx 0.05$ it stabilises around $h \approx 0.04$, also see *Appendix 1—figure 5*) HGT, the rescuable gene, and the SGEs are evolutionarily maintained. When the influx of SGEs is stopped, the cells (and their beneficial gene) stably coexists with these genetic parasites.

Strikingly, if we introduce SGEs whose fitness penalty is greater than the benefit of the gene ('strong SGEs', $\beta = 0.04$), we also observe the coexistence of cells, rescuable genes, and SGEs. By looking at the initial invasion dynamics (*Figure 6B*), we can see that these strong SGEs cannot rise to

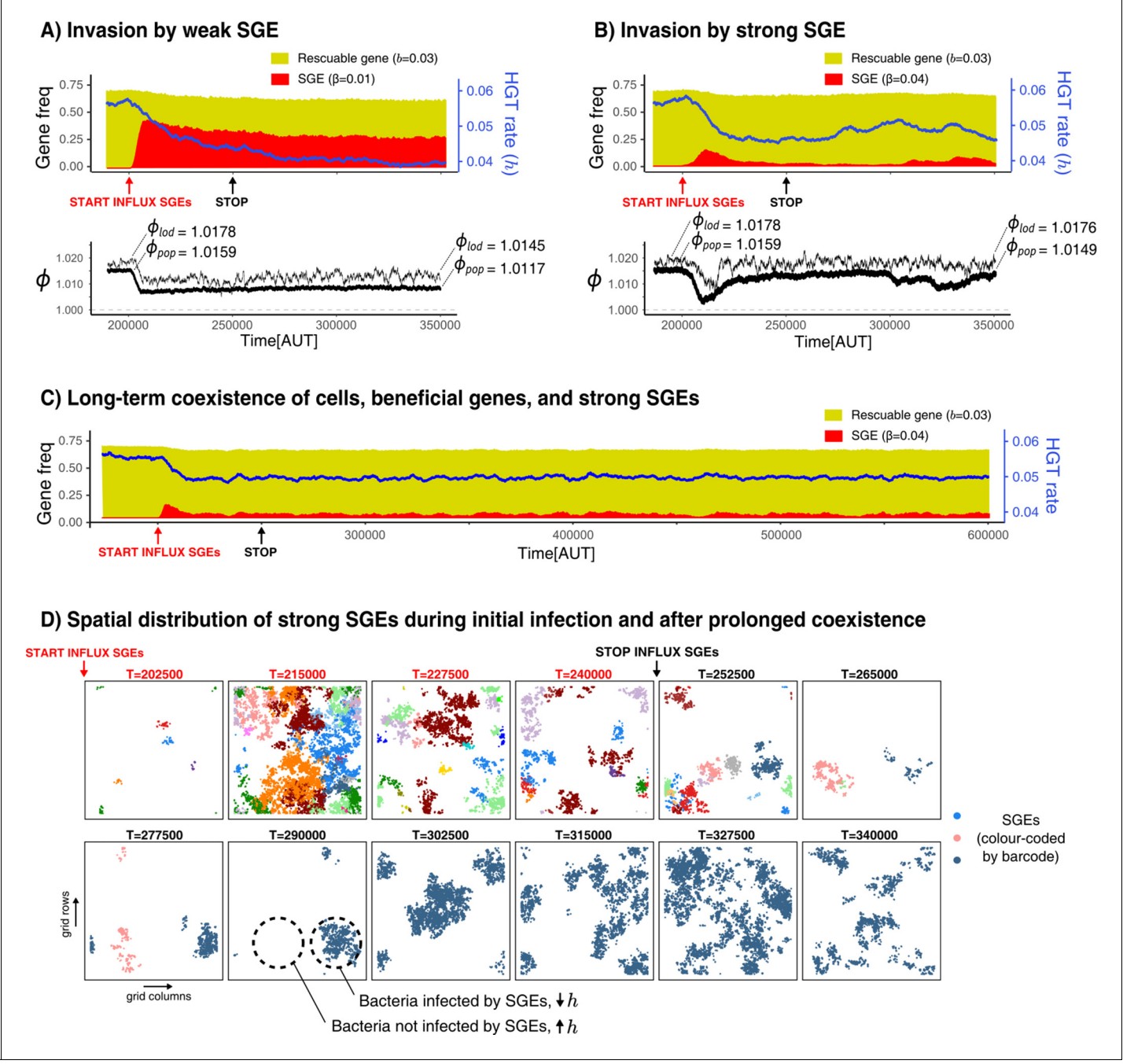

**Figure 6.** Selfish genetic elements (SGEs) can invade and stably coexist with their host cell. (A-B) Temporal dynamics for a population that has evolved to rescue a slightly beneficial gene ($b = 0.03$) invaded by a weak SGE (A, $\beta = 0.01$) and a strong SGE (B, $\beta = 0.04$) respectively. The blue line indicates the rate of HGT (i.e. DNA uptake) as evolved by the host cells. In the bottom graphs, the thick black line is the average growth rate of the population ($\phi_{pop}$), and the thin black line is $\phi_{lod}$, the average growth rate of individuals along the line of descent (up to 250 generations ago) . The $\phi_{pop}$ and $\phi_{lod}$ that are annotated with the dashed lines are the average of the first/final 200 generations. (C) This panel shows the long-term coexistence of cells, beneficial genes, and a strong SGEs ($\beta = 0.04$). (D) Panels show the spatial distribution of SGEs (coloured by their unique barcodes). The top row shows this during the invasion (open ecosystem) and the bottom row shows this during prolonged coexistence (closed ecosystem). Note that the empty sites (white) only indicate the absence of SGEs, not of bacterial cells, which are instead present in every grid point. Parameters used: $h$-parameters and frequency of carriers as evolved from **Figure 5**, $i = 5 \cdot 10^{-3}$, $m = 0.05$, $l = 0.02$, $c = 0.1$, $i = 1 \cdot 10^{-5}$, $i_{start} = 200,000$, $i_{stop} = 250,000$, $n = 400$ (i.e. $N = 400^2$).

very high frequencies. As the hosts evolve lower rates of DNA uptake, these genetic parasites are pushed to very low frequencies. However, the reduced threat of genetic parasites causes the host cells to once again increase their rates of DNA uptake, leading to a secondary outbreak of SGEs (*Figure 6B*, from T = 300,000 onwards). It is interesting to note that, while the population growth rates ($\phi_{pop}$) clearly decrease due to this second infection, the growth rates along the line-of-descent ($\phi_{lod}$, see Materials and methods) remains largely unaffected. Thus, while a sub-set of the population has been infected, individuals in this infected strain will not be amongst the long-term ancestors. Counter-intuitively, strong SGEs only have a minor impact on bacterial growth rates, as they are only retained at very low frequencies. Weaker SGEs instead impose a significant burden on the population by rising to much higher frequencies. Indeed, when SGEs are very costly, the population purges them entirely by evolving lower rates of DNA uptake (see *Appendix 1—figure 5C*). Finally, note how stopping the influx of SGEs does not impact the long-term coexistence of cells, beneficial genes, and these strong SGEs ( *Appendix 1—figure 5C*).

To better understand the co-evolutionary process between SGEs and bacteria engaging in HGT of rescuable genes, *Figure 6D* shows long-term dynamics of barcoded SGEs in this spatial system. Although a diverse set of SGEs are initially discovered in parallel (coloured by their unique barcode), eventually only a single barcode remains after the influx of SGEs is stopped. Moreover, it can also be seen how SGEs are either locally abundant, or entirely absent. Thus, spatially separated strains of bacteria experience opposing selection pressures for taking up DNA. Lower rates of uptake are favoured in the presence of these strong SGEs, but higher rates of uptake are favoured when these genetic parasites have (locally) died out. Indeed, this heterogeneity of SGEs is crucial for the strong SGEs to persist, as well-mixed populations can only retain weaker SGEs (see *Appendix 1—figure 5*). Interestingly, we also found that strong SGEs failed to persist when HGT was too localised (e.g. only between neighbouring cells), as the SGEs then could not escape to a new pool of hosts that have high rates of DNA uptake (*Appendix 1—figure 6*). We conclude that, in a spatially structured population, strong SGEs can stably coexist in a bacterial population which maintains its ability to engage in HGT to 'rescue' rescuable genes.

## Discussion

We have studied the balance between the advantages and disadvantages of HGT by modelling a simple bacterial population undergoing the uptake of genes from a shared DNA pool. Our analysis shows that we can categorise slightly beneficial genes based on whether genes are lost from the population without HGT, and whether HGT of these genes can improve the population growth rate. This results in five distinct gene classes: (i) *indispensable genes*, that readily persist within the population and for which HGT is therefore always deleterious, (ii) *enrichable genes* which are not lost from the population without HGT, but moderate rates of HGT are adaptive, (iii) *rescuable genes* which are lost from the population without HGT, but can be rescued by HGT which improves population growth rates, and (iv) *unrescuable genes*, that are also lost from the population without HGT, but recovering them with HGT does not improve population growth rates, and (v) *selfish genetic elements*, genes that confer a fitness penalty, but can persist within the population with HGT. We further investigated if HGT of these respective gene classes is an evolutionarily stable strategy, and if the bacterial cells can evolve to engage in HGT de novo. We found that horizontal transfer of enrichable and rescuable genes is indeed a evolutionarily stable strategy, but can only evolve from scratch for enrichable genes. The evolution of HGT to 'rescue' a rescuable gene faces a problem under well-mixed conditions: HGT is required for the gene to persist, but sufficient carriers of the gene are necessary to evolve HGT. By modelling this process in a spatially structured population, we show that HGT can nevertheless evolve for these rescuable genes. As carriers of the gene can be locally abundant, emergent communities form that locally retain the gene via HGT, therewith slowly outgrowing other individuals. Finally, we show that once stable transferring communities have evolved, selfish genetic elements (SGEs) can stably coexist with the bacterial population and the beneficial genes. In spite of these genetic parasites exploiting the host's ability to transfer, HGT is evolutionarily maintained, providing a doorway to the co-evolutionary process between bacteria and SGEs.

Here, we have used a simple model which ignores many of the complexities of natural bacterial populations, for example by assuming a constant environment and studying the impact of HGT one gene at a time. In reality, microbial ecosystems experience many changing selection pressures.

Besides fluctuations in the available resources, traits like antimicrobial resistance, toxin production, and cooperative behaviour may confer large fitness benefits only under specific ecological circumstances (*Riley and Wertz, 2002*; *Cordero et al., 2012*; *Vogwill and MacLean, 2015*; *Gerardin et al., 2016*; *Hehemann et al., 2016*; *Dimitriu et al., 2019*). Moreover, when considering the impact of HGT, we should also consider its implications for the genome as a whole, rather than only considering single genes. To take into account the impact of HGT on the whole genome, it is important to consider the possibility of differential gene mobility, where genes associated with MGEs may transfer much more frequently than other genes (*Rankin et al., 2011*). However, we should then also take into account selection pressures that allow MGEs to have both mutualistic and parasitic relationships with their host (*Harrison et al., 2015*). Although our extremely simple model ignores all the above intricacies, it retrospectively shines light on dynamics observed in our earlier eco-evolutionary modelling of antagonistic interactions (*van Dijk and Hogeweg, 2016*). In this multi-level model, which included changing selection pressures and the co-evolution of cells and mutualistic/parasitic genes, we showed that rarely beneficial toxin genes evolved to transfer much more frequently than their corresponding resistance factors. Despite the complex eco-evolutionary dynamics in this model, we can in hindsight understand how toxin genes evolved to correspond to 'rescuable' genes, and the resistance genes evolved to be 'indispensable'. However, the rescuable toxin genes in this model experienced alternating selection pressures, only sporadically being highly beneficial for their hosts. Therefore, it remained unclear whether the ability of HGT to rescue genes required such alternating selection pressures, which was suggested to be important in earlier studies (*Bergstrom et al., 2000*; *Rankin et al., 2011*). Here, our simple model has revealed that bacteria may also benefit from preferentially mobilising genes that are, constantly, slightly beneficial.

We have studied the impact of HGT by considering bacterial transformation as the mechanism by which genes transfer, that is by assuming (i) engaging in HGT is continuously costly and not only when it is successful, (ii) all carrier cells can act as a donor regardless of their ability to take up DNA themselves, and (iii) the ability for HGT is determined by the recipient cell. However, note that similar gene classes and comparable dynamics are observed when modifying these assumptions in our model (see *Appendix 1—figure 1*; *Appendix 1—figure 2*; *Appendix 1—figure 3*), and that some of our key results may therefore hold for different mechanisms of HGT like conjugation. Indeed, there are some interesting parallels with earlier modelling work on plasmid persistence by *Bergstrom et al., 2000*. On the one hand, this study shows that if genes carried on plasmids are sufficiently beneficial, HGT is not necessary, and plasmid-bearing cells are outcompeted by plasmid-free cells that have integrated the beneficial trait into the bacterial chromosome. These dynamics resemble those of the indispensable genes in our model, where transfer is not adaptive as the trait readily persists without HGT. On the other hand, Bergstrom et al. show that plasmids that are intermittently favoured can be retained by shuttling them back and forth between strains or species, reminiscent of the dynamics we have revealed for rescuable genes. However, there are of course some very important differences between conjugation and transformation, where plasmids for example confers other burdens on the host (*San Millan and Craig maclean, 2019*), cells require direct contact to conjugate, and the F-factor necessary for conjugation can itself reside either on a plasmid or on the bacterial chromosome. A promising direction for future modelling therefore is to more directly compare these two distinct mechanisms of HGT, so that we may one day learn to predict which genes will spread in an evolving microbial population.

Besides investigating the impact of HGT for a range of different fitness effects, we have also shown how spatial structure is a key component for the emergence of HGT of rescuable genes. Both conjugation and transformation have indeed been observed to occur more frequently in biofilms than in well-mixed cultures (*Madsen et al., 2012*), and plasmids have furthermore been shown to be more persistent in biofilms (*Stalder et al., 2020*). On the other hand, spatial structure can slow down adaptation in asexual populations because individuals are mostly competing with their related conspecifics (*Gordo and Campos, 2006*; *Habets et al., 2007*; *Chacón and Harcombe, 2019*). Relatedness has indeed been shown to be an important factor in stabilising HGT, for example of cooperative traits (*Mc Ginty et al., 2011*). Our model shows that, also without explicitly taking cooperation into account, HGT can only evolve in 'gene-sharing' communities which emerge in by local reproduction in spatially structured populations. Thus, not only are relatedness and spatial structure necessarily intertwined, they are crucial for the rare 'nucleation events' that initiates evolution towards increased rates of HGT. Intriguingly, similar nucleation events have been observed in origin of life

studies (*Wu and Higgs, 2012*) and models of microbial antagonistic interactions (*Kotil, 2018*). These types of emergent evolutionary transitions highlight how studying evolution under well-mixed conditions, and one mutant at a time, can be highly misleading. Studying biological systems in a spatial context will help us to better understand which eco-evolutionary outcomes are accessible, and maintainable, by evolution.

### Horizontal gene transfer: rescue or catastrophe?

In nature, HGT can happen through a variety of mechanisms that each have their own potential advantages and disadvantages for the host cell (*Vogan and Higgs, 2011*; *Baltrus, 2013*). Bacteria do not always have full control over the rates at which HGT happens, especially when considering it as a side-effects of other processes (*Redfield, 2001*). However, it remains an intriguing question under which specific circumstances bacteria benefit from HGT, whether it is a side-effect or not. By abstracting away from the different mechanisms of HGT, and what it means for a gene to be 'beneficial', we have revealed the conditions under which HGT is an adaptive trait for the host cells. In a similar spirit, earlier modelling by *Vogan and Higgs, 2011* has shown that HGT can be adaptive with respect to genes that are frequently lost. However, in their work, natural selection eventually favoured improved replication accuracy, therewith decreasing the advantage of HGT. Other models have shown that HGT is beneficial to mitigate the effects of *Muller, 1964* by decreasing assortment load (*Takeuchi et al., 2014*; *Vig-Milkovics et al., 2019*), analogous to the impact of sex and recombination on the balance between drift and selection (*Lynch et al., 1995*; *Schultz and Lynch, 1997*; *Lynch et al., 2016*; *Vos et al., 2019*). Our work complements these aforementioned studies by showing that, however low the rate of gene loss may be, there may always be a class of slightly beneficial traits for which HGT is adaptive and evolvable. Although genes with such small fitness effects are very hard to detect experimentally (*Bataillon, 2000*; *Wiser and Lenski, 2015*), our model is a proof of principle that HGT may play a key role in preventing the loss of these genes, which may explain the differential rates of HGT as observed in the data (*Nogueira et al., 2009*; *Rankin et al., 2011*; *Madsen et al., 2012*; *Novick and Doolittle, 2020*). With the upswing and improvement of experimental techniques like Hi-C metagenomics (*Beitel et al., 2014*; *Burton et al., 2014*) and DNA barcoding (*Blundell and Levy, 2014*; *Nguyen Ba et al., 2019*), we will soon have more insights into the eco-evolutionary dynamics of small-effect mutations (*Li et al., 2018*; *Lerner et al., 2019*) and accessory genes (*Quistad et al., 2019*; *Yaffe and Relman, 2020*), and we may learn when HGT can come to rescue a microbial population, and when it may be nothing more than a catastrophe.

## Materials and methods

### General overview

In this work, we study the dynamics of bacteria undergoing HGT of slightly beneficial genes and Selfish Genetic Elements (SGEs). We do this by modelling the same processes with gradually increasing complexity, starting from simple Ordinary Differential Equations (ODEs), and then evaluating the same dynamics in an Invididual-based Model (IBM). A graphical representation of these models is found in the main text (*Figure 1*). The models consider the competition between cells of two types: carrier cells ($C$) that carry a gene, and non-carrier cells ($N$). When carrier cells contain a beneficial gene (i.e. it is a beneficial trait), they grow faster than the non-carrier cells ($N$). However, carriers may lose this beneficial gene with a fixed rate $l$. Both cell types take up DNA with rate $h$, which comes with a cost $c$. This cost is equal for both cell types, reflecting for example the costs of expressing and operating the DNA uptake machinery, other metabolic burdens of the physiological state of natural competence, or the risks associated with taking up foreign DNA. Proportional to the density of available carrier cells, non-carriers can be transformed back into a carrier cell by means of 'additive' HGT. Both models use a chemostat assumption, where cells wash out at a rate proportional to the rate of growth, ensuring a constant population size in steady state.

### ODE model(s)

By modelling the dynamics described above by means of ODEs, we assume a well-mixed population of cells that compete according to all-against-all dynamics (i.e. mass-action). Our equations describing the density of carrier ($C$) and non-carrier ($N$) cells are given in *Equation 2*, where $b$ is the benefit

of the carried gene (or burden if $b<0$), $l$ is the rate of gene loss, $h$ is the rate at which cells engage in HGT, $c$ is the continuous cost for engaging in HGT, and HGT transforms a non-carrier into a carrier when they interact ($hCN$). This cost for HGT ($c$) is equal for both cell types, meaning that whatever the costs may entail, we assume they are continuously payed. Finally, the total amount of growth ($\phi$) is subtracted from both populations, meaning that the population density in steady state is always 1.

$$\frac{\mathrm{d}C}{\mathrm{d}t} = \underbrace{(1-ch+b)C}_{\text{reproduction of C}} - \underbrace{lC}_{\text{gene loss}} + \underbrace{hCN}_{\text{HGT}} - \underbrace{\phi C}_{\text{chemostat}}$$

$$\frac{\mathrm{d}N}{\mathrm{d}t} = \underbrace{(1-ch)N}_{\text{reproduction of N}} + \underbrace{lC}_{\text{gene loss}} - \underbrace{hCN}_{\text{HGT}} - \underbrace{\phi N}_{\text{chemostat}}$$

$$\phi = \underbrace{(1-ch+b)C}_{\text{total growth of C}} + \underbrace{(1-ch)N}_{\text{total growth of N}}$$

$$C+N = 1 \text{ (constant population size, ensured by chemostat assumption.)}$$

(2)

From the above model (*Equation 2*), we derived how the population growth rate ($\phi$) depends on both $b$ and $h$ (see *Equation 1* in the main text), which shows the conditions under which HGT improves the total growth rate of the population. To analyse whether or not HGT could evolve, we extended the two-variable ODE model above (of cells with the same $h$) to a four-variable ODE model (of two species with a different $h$, see *Figure 1B* and *Equation 3* below). We use this extension to study whether or not a species with HGT ($C^+$ and $N^+$, $h>0$) could invade upon a species without HGT ($C^-$ and $N^-$, $h=0$), and *vice versa* (see Supplementary material for full analysis). Finally, we also extended the ODE model to study the impact on growth rates for cells that engage in HGT of both a beneficial gene and a Selfish Genetic Element (SGE), which can be found in the Supplementary Material.

$$\frac{\mathrm{d}C^-}{\mathrm{d}t} = (1+b)C^- - lC^- - \phi C^-$$

$$\frac{\mathrm{d}N^-}{\mathrm{d}t} = N^- + lC^- - \phi N^-$$

$$\frac{\mathrm{d}C^+}{\mathrm{d}t} = (1+b-ch)C^+ - lC^+ + hN^+(C^- + C^+) - \phi C^+$$

$$\frac{\mathrm{d}N^+}{\mathrm{d}t} = (1-ch)N^+ + lC^+ - hN^+(C^- + C^+) - \phi N^+$$

$$\phi = (1+b)C^- + N^- + (1+b-ch)C^+ + (1-ch)N^+$$

(3)

## Individual-based model

The individual-based model (IBM) describes the same dynamics as the ODE models, but differs in some important aspects. Firstly, individuals are discrete entities that live on a 2D grid, and reproduce locally. This allows us to study the model with and without spatial pattern formation by modifying the rate at which cells mix. When mixing is disabled or very limited, a spatially structured population like that of a biofilm will form, while an increased amount of cellular mixing will approximate a well-mixed culture. Under well-mixed conditions, individuals will interact with random individuals in the population (approximating the all-against-all dynamics of the ODEs), while individuals will interact mostly with their conspecifics in case of the biofilm. We explicitly define a competition range (focal cell plus its eight neighbouring grid points) and a HGT range (all cells within distance $t$) which determine smaller samples of the total population with which individuals can interact. Each individual (potentially) has its own $h$-parameter, allowing us to study the evolution of HGT in an eco-evolutionary context (see implementation of mutations below). As we primarily focus on the question if cells benefit from taking up genes from their environment or other cells, we assume that the $h$-parameter of the recipient cell determines the probability of HGT. While we tested if HGT could evolve de novo by studying the invasion critereon for $HGT^+$ in the ODE model, we study the de novo evolution of HGT in the IBM by continuously introducing carriers at a low rate. With a rate $f$, genes with benefit $b^*$ are (re)discovered, allowing us to study how and if newly discovered genes/selfish elements spread through the population. Finally, note that processes such as gene loss, HGT, and competition are no longer deterministic like in the ODEs, but implemented as events that can stochastically happen at each simulated time step. To ensure the chance-events in the IBM (reproduction, HGT, gene

loss) accurately represent the rates as used in the ODE, all probabilities were multiplied by a small constant $\Delta T = 0.1$.

## Updating grid points

All grid points $i,j$ in the IBM contain a single cell which can be a carrier or non-carrier ($b_{i,j} = b$ for carriers, $b_{i,j} = 0$ for non-carrier), which can carry a SGE ($\beta_{i,j} = \beta$ for SGE infected cells, $\beta_{i,j} = 0$ for uninfected cells), and has an individual HGT-parameter $h_{i,j}$. At each time step, local reproduction happens in each grid point $i,j$ by drawing a random individual from the Moore (nine cells) neighbourhood and letting it reproduce with a probability proportional to its growth-rate $\varphi_{i,j}$:

$$\varphi_{i,j} = 1 + b_{i,j} - \beta_{i,j} - ch_{i,j} \qquad (4)$$

When reproduction happens, the winner cell replicates and replaces the cell in grid point $i,j$. This newborn cell is an exact copy of the mother cell. Next, all cells are also updated to include the processes of stochastic gene loss with rate $l$, HGT with rate $h_{i,j}$, and gene/SGE discoveries based on the influx-rate $f$. Finally, with a small probability $u$, the HGT rate of any individual can mutate, where a cell uniformly samples a new parameter between $h_{i,j} - m$ and $h_{i,j} + m$.

## IBM growth rates

With respect to growth rates, the simulated IBM model does not only track the average growth rate of all cells in the population ($\phi_{pop}$), but also tracks the growth rate of the line of descent that gave rise to the current population ($\phi_{lod}$). While $\phi_{pop}$ is comparable to $\phi$ in the ODE-model, $\phi_{lod}$ gives us insights into how the long-term ancestors are impacted by HGT. We also measure the average competitive advantage of carrier cells ($\sigma_c$) by calculating, for each carrier cell, how much higher *varphi* is than that of the eight neighbouring individuals. When all competitors of a also carrier cells, this value is 0. When all competitors are non-carriers, this value approaches $b$ is the average of these values.

## IBM barcoding

We tag all influxed genes and SGEs with a unique identifier, allowing us to visualise how genes/SGEs spread through the population (analogous do DNA barcoding [*Blundell and Levy, 2014*; *Levy et al., 2015*; *Nguyen Ba et al., 2019*; *Lerner et al., 2019*]). These barcodes also allow us to

**Table 1.** Description of parameters used in the models.

| Parameter (general) | Description |
| --- | --- |
| Gene loss ($l$) | Rate at which carrier cells lose the beneficial gene |
| HGT rate ($h$) | Rate at which non-carriers are transformed into carriers (when interacting with carrier cells) |
| Benefit of gene ($b$) | Growth rate benefit for carrier cells (or penalty for negative b) |
| Costs of HGT ($c$) | Growth rate penalty for the rate of HGT |

| Parameter (IBM only) | Description |
| --- | --- |
| Grid size ($n$) | The simulation is done on a square grid of n x n cells |
| Mixing rate ($d$) | Every time step, the grid is mixed $d$ times using the Margolus Diffusion algorithm (*Toffoli and Margolus, 1987*). Alternatively, the population was *well-mixed* by assigning new positions at random every time step. |
| Competition range ($s$) | Sub-population of s x s cells surrounding focal grid point that compete for reproduction |
| HGT distance ($t$) | Sub-population of t x t cells surrounding focal grid point from which a random potential donor is sampled for HGT |
| Influx genes ($f$) | A small probability for any cell to discover a gene de novo |
| Benefit of influxed gene ($b*$) | Growth rate benefit for carrier cells (or penalty for negative b) |
| Influx SGEs ($i$) | A small probability for any cell to be infected by an SGE de novo |
| Fitness penalty of SGE ($\beta$) | The fitness penalty imposed by the SGE |
| Mutation rate ($u$) | Chance of mutating the evolvable HGT-rate |
| Mutation step ($m$) | Uniform step size of mutations |

investigate whether or not these genes are continuously rediscovered, or form long lineages of genes that persist within the population.

## Parameters used

Throughout most of this study, the gene loss $l$ was set to 0.02 and the cost for HGT was set to $c = 0.2$. In general, our results do not depend on the absolute value of these two parameters. Instead, our results depends on the relationship between these two parameters, meaning that for lower (arguably more realistic) rates of gene loss, the relevant parameter range will still be extensive if the costs for HGT are also lower. If the costs are very high, the range where HGT is adaptive (i.e. enrichable and rescuable genes) becomes much more narrow, but is nevertheless retained (i.e. the gene classes discussed in *Figure 2* simply shift to lower values of $b$). As these costs may entail a wide variety of different burdens on the cell (operating and expressing DNA pumps, diverting resources to the physiolocial state of natural competence, the burden of expressing redundant gene copies, chromosome disruptions, cytotoxicity, and the risks of SGEs), we do not know quantitative information to judge the relevant value for this cost parameter. Instead, we argue that our model instead gives conceptual clarifications and demonstrations of novel possibilities. Parameters such as the benefit ($b$), the HGT-rate ($h$), the amount of mixing ($d$), and the HGT distance ($t$) have been extensively swept, as discussed in the main text/Supplementary Material. In these cases, the used parameters are given in the captions of the relevant figures. When comparing the IBM with the ODE models (e.g. occurrence of gene classes), evolution of $h$ was disabled ($u = 0.0$). For the de novo evolution of HGT, the initial population consisted only of *non-carrier* cells, but genes fluxed in at a low rate ($f = 5 \cdot 10^{-6}$), while the initial level of HGT ($h = 0.0$) was allowed to evolve with $f = 5 \cdot 10^{-5}$ with a uniform step size of $m = 0.05$. Finally, when testing whether HGT could be maintained, no influx of genes was present ($f = 0.0$), but the initial population consisted of carrier-cells that already engage in HGT ($h = 0.05$, see supplementary material for evolved rates of HGT). All experiments in the IBM with Selfish Genetic Elements were done with slightly lower costs ($c = 0.1$), to compensate for the extra costs imposed by these genetic parasites.

All the important parameters of our models are summarised in *Table 1*.

## Software used

The analytical model was numerically analysed using grind.R by R.J. de Boer (http://tbb.bio.uu.nl/rdb), an R script that uses the deSolve R-package (*Soetaert et al., 2010*). The simulated model was implemented in Cash (Cellular Automaton simulated hardware) version 2.1, an free and easy-to-use library to make simple spatially explicit simulations (originally created by R.J. de Boer and A.D. Staritsk, further developed by Nobuto Takeuchi and Bram van Dijk). Visualisation of both models was done in R using ggplot (*Wickham, 2016*) and plotly (*Inc PT, 2015*). Simulations were run in Linux Ubuntu 16.04 LTS using GNU parallel (*Tange, 2018*).

Both the R-scripts for ODE analysis and the IBM code implemented in C, are available online https://github.com/bramvandijk88/HGT_Genes_And_SGEs (*van Dijk, 2020*; copy arhived at https://github.com/elifesciences-publications/HGT_Genes_And_SGEs).

## Acknowledgements

This work was supported by the European Commission 7th Framework Programme (FPFP7-ICT-2013.9.6 FET Proactive: Evolving Living Technologies) EvoEvo project (ICT-610427) and the Human Frontier Science Program (RGY0072/2015).

## Additional information

### Funding

| Funder | Grant reference number | Author |
| --- | --- | --- |
| Seventh Framework Programme | ICT-610427 | Bram van van Dijk<br>Paulien Hogeweg |
| Human Frontier Science Program | RGY0072/2015 | Hilje M Doekes |

The funders had no role in study design, data collection and interpretation, or the decision to submit the work for publication.

## Author contributions
Bram van Dijk, Conceptualization, Resources, Data curation, Software, Formal analysis, Validation, Investigation, Visualization, Methodology, Writing - original draft, Project administration, Writing - review and editing; Paulien Hogeweg, Conceptualization, Supervision, Funding acquisition, Methodology, Writing - review and editing; Hilje M Doekes, Formal analysis, Validation, Investigation, Methodology, Writing - review and editing; Nobuto Takeuchi, Conceptualization, Formal analysis, Supervision, Validation, Methodology, Writing - review and editing

## Author ORCIDs
Bram van Dijk https://orcid.org/0000-0002-6330-6934
Hilje M Doekes http://orcid.org/0000-0002-6360-5176

## Decision letter and Author response
Decision letter https://doi.org/10.7554/eLife.56801.sa1
Author response https://doi.org/10.7554/eLife.56801.sa2

---

# Additional files
## Supplementary files
• Transparent reporting form

## Data availability
All data are either mathematical or computationally generated, and therefore easily reproduced. All scripts and programs to so do are publicly available on GitHub (https://github.com/bramvandijk88/HGT_Genes_And_SGEs copy arhived at https://github.com/elifesciences-publications/HGT_Genes_And_SGEs). For Figure 2 and 3 we used the analytical model. To (numerically) reproduce our results, use the Rscripts provided in the repository. For Figure 4, 5 and 6 we used the individual-based model. This was implemented in C, and can be run with simple command-line options (readme file found in the zip).

---

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

## Appendix 1

## Supplementary material

This supplementary material includes the mathematical derivations of the results discussed in the main text and some extra insights and figures. The source code material to reproduce the numerical simulations we have done (both in the main text and in this supplementary material), is available online (https://github.com/bramvandijk88/HGT_Genes_And_SGEs).

## Part I: Mathematical analyses

### Equilibria and population growth rate of a single population

As described in the main text, we consider a population of cells that either carry or do not carry a gene. The dynamics of the density of carriers ($C$) and non-carriers ($N$) are described by:

$$\frac{dC}{dt} = \underbrace{(1 - ch + b)C}_{\text{reproduction of C}} - \underbrace{lC}_{\text{gene loss}} + \underbrace{hCN}_{\text{HGT}} - \underbrace{\phi C}_{\text{chemostat}} \tag{5}$$

$$\frac{dN}{dt} = \underbrace{(1 - ch)N}_{\text{reproduction of N}} + \underbrace{lC}_{\text{gene loss}} - \underbrace{hCN}_{\text{HGT}} - \underbrace{\phi N}_{\text{chemostat}} \tag{6}$$

$$\phi = \underbrace{(1 - ch + b)C}_{\text{total growth of C}} + \underbrace{(1 - ch)N}_{\text{total growth of N}} \tag{7}$$

$$C + N = 1 \text{ (constant population size, ensured by chemostat assumption.)} \tag{8}$$

### Equilibria and their stability

The equilibria of *Equations 5–8* are found by solving $\frac{dC}{dt} = \frac{dN}{dt} = 0$.

$$Let \; \frac{dC}{dt} = (1 + b - ch)C - lC + hNC - \phi C = 0.$$

Then either $C = 0$, or $\phi = 1 + b - ch - l + hN$ and

$$
\begin{aligned}
& 1 + b - ch - l + hN = (1 + b - ch)C + (1 - ch)N \\
\Longleftrightarrow \; & 1 + b - ch - l + h(1 - C) = (1 + b - ch)C + (1 - ch)(1 - C) \\
\Longleftrightarrow \; & 1 + b - ch - l + h = (1 + b - ch + h - 1 + ch)C + 1 - ch \\
\Longleftrightarrow \; & b - l + h = (b + h)C \\
\Longleftrightarrow \; & C = \frac{b - l + h}{b + h} = 1 - \frac{l}{b + h}
\end{aligned}
$$

Using $C + N = 1$, we find that the system has two equilibria:

$$\text{equilibrium(i)}: \quad C^* = 0, \quad N^* = 1, \tag{9}$$

$$\text{equilibrium(ii)}: \quad C^* = 1 - \frac{l}{b + h}, \quad N^* = \frac{l}{b + h} \tag{10}$$

Next, we study under what conditions the gene can persist in the population described by *Equations 5–8*. Note that this is equivalent to asking when equilibrium (i) is unstable, that is when the carrying cells ($C$) can invade on a resident population of non-carrying cells ($N$) at carrying capacity. When the system is in equilibrium (i), $C^* = 0$, $N^* = 1$, and $\phi^* = (1 - ch)$. The dynamics of the carrying cells can then be approximated by

$$\frac{\mathrm{d}C}{\mathrm{d}t} \approx (1+b-ch-l+hN^*-\phi^*)C = (1+b-ch-l+h-(1-ch))C = (b+h-l)C,$$

and the carrying cells can invade if $\frac{\mathrm{d}C}{\mathrm{d}t} > 0$, that is if

$$b+h-l > 0. \tag{11}$$

From *Equation 11,* we can conclude that genes which yield a sufficient growth rate benefit to overcome the loss rate ($b>l$) do not need HGT in order to persist in a population. Slightly beneficial genes, however, only persist when $h>(l-b)$. HGT, serving as a plausible 'back-mutation', prevents the eventual loss of such a gene from the population.

## Population growth rate $\phi$ in steady state as a function of HGT rate $h$

Even though we have shown above that some genes can only persist in a population at sufficiently high rates of HGT, the survival of these genes does not necessarily imply that HGT also improves the actual growth rate of the population under these conditions, as the model also assumes a cost for higher rates of HGT. To gain better insight into when HGT improves the steady state growth rate, we will next consider how the population growth rate $\phi$ depends on $h$.

The population growth rate in steady state, $\phi^*$, is given by:

$$\phi^*(h) = (1+b-ch)C^* + (1-ch)N^* \tag{12}$$

$$= \begin{cases} 1-ch & \text{if } h \leq (l-b) \text{ (gene cannot persist)}; \\ 1-ch+b-\frac{bl}{b+h} & \text{if } h>(l-b) \text{ (gene persists)}. \end{cases} \tag{13}$$

To determine the effect of the rate of HGT, $h$, on the steady state population growth rate $\phi^*$, we differentiate *Equation 13* with respect to $h$:

$$\frac{\partial \phi^*}{\partial h} = \begin{cases} -c & \text{if } h \leq (l-b); \\ -c + \frac{bl}{(b+h)^2} & \text{if } h > (l-b). \end{cases} \tag{14}$$

As long as $h<(l-b)$, $\frac{\partial \phi^*}{\partial h} = -c < 0$ and an increase in HGT rate $h$ will decrease the population growth rate at steady state $\phi^*(h)$. For, $h>(l-b)$, the population growth rate $\phi^*$ might however have a local optimum, which we can find by setting $\frac{\partial \phi^*}{\partial h}$ to 0:

$$\frac{bl}{(b+h)^2} - c = 0$$

$$\iff (b+h)^2 = \frac{bl}{c}$$

from which we can solve

$$h_{\mathrm{opt}} = \sqrt{\frac{bl}{c}} - b \tag{15}$$

Note that this optimum is only obtained in the function $\phi^*(h)$ if $h_{\mathrm{opt}} > (l-b)$:

$$\sqrt{\frac{bl}{c}} - b > l - b \tag{16}$$

$$\iff \frac{bl}{c} > l^2 \tag{17}$$

$$\Longleftrightarrow b>lc. \tag{18}$$

(This is the same condition found when solving $\frac{\partial \phi^*}{\partial h}>0$ at $h=(l-b)$)

Furthermore, since $h$ is the rate of HGT, we are only interested in positive values of $h$ if

$$\sqrt{\frac{bl}{c}}>b \tag{19}$$

$$b<\frac{l}{c}. \tag{20}$$

Under the conditions of **Equations 18 and 20**, the second derivative of $\phi^*$ to $h$ is

$$\frac{\partial^2 \phi^*}{\partial h^2}=\frac{-2bl}{(b+h)^3},$$

which is negative if the parameters $b$ and $l$ are $\geq 0$. Hence, when $\phi^*(h)$ has an optimum for a positive HGT rate $h_{\text{opt}}$, this local optimum is a maximum. The growth rate in this local maximum is larger than the growth rate at $h=0, \phi^*(0)=1$, if

$$\phi^*(h_{\text{opt}})=1-ch_{\text{opt}}+b-\frac{bl}{b+h_{\text{opt}}}>1 \tag{21}$$

$$\Longleftrightarrow 1+cb-\sqrt{bcl}+b-\frac{bl}{\sqrt{\frac{bl}{c}}}>1 \tag{22}$$

$$\Longleftrightarrow b(1+c)-2\sqrt{bcl}>0 \tag{23}$$

$$\Longleftrightarrow b>\frac{4lc}{(1+c)^2}. \tag{24}$$

Summarising, the population growth rate at equilibrium, $\phi^*$, decreases linearly with the risks $ch$ when $h<(l-b)$ due to the costs of HGT (see **Equation 13**). Under these conditions, the growth rate does not depend on $b$ because the gene cannot persist in the population. When $h>(l-b)$, the gene does persist within the population, resulting in an extra term $b-\frac{bl}{b+h}$ in the growth rate $\phi^*(h)$. This extra term approaches a maximal benefit of $b$ for high values of $h$. The burden of HGT $ch$ will however eventually outweigh this benefit for increasing rates of HGT. A (local) optimal rate of HGT can found at $h_{\text{opt}}=\sqrt{bl/c}-b$, as long as $b>lc$. This optimal HGT rate is greater than 1, meaning that HGT improves the population growth rate at steady state, if the genes have a minimal benefit (see **Equation 24**). However, when the benefit is too large ($b>l/c$), the optimal HGT rate becomes $h_{\text{opt}}<0$. As negative values for HGT are biologically unsound, HGT never improves the population growth rate in steady state for genes with such a high fitness benefit. Following these derivations, genes can be divided in different classes based on the value of the fitness benefit $b$ and the consequent effect of HGT on the population growth rate at steady state (see main text and **Figure 2**):

## Selfish Genetic Elements (SGEs) ($b<0$)

Carrying the gene confers a fitness cost. Increasing HGT-rates only lower the equilibrium population growth rate $\phi^*$.

## Unrescuable genes ($b<l$ and $b<\frac{4lc}{(1+c)^2}$)

Genes confer a small fitness benefit, but this benefit is too small to overcome gene loss. Furthermore, no positive HGT rate $h$ improves the population growth rate $\phi^*(h)$ over the population growth rate in the absence of HGT ($\phi^*(0)=1$).

### Rescuable genes ($\frac{4lc}{(1+c)^2} < b < l$)

Genes confer a small fitness benefit and cannot persist in a population in the absence of HGT, but can be rescued by a sufficiently high HGT rate ($h > (l - b)$). For some HGT rate $h_{\text{opt}} > 0$ the equilibrium growth rate $\phi^*(h) > 1$, indicating that HGT can improve the growth rate of the population.

### Enrichable genes ($l < b < l/c$)

Genes confer a sufficient fitness benefit to persist in a population in the absence of HGT. HGT can however improve the equilibrium population growth rate $\phi^*(h_{\text{opt}})$.

### Indispensable genes ($b > l/c$)

Genes confer a large fitness benefit and can persist in a population in the absence of HGT. HGT furthermore does not improve the equilibrium population growth rate.

## Evolutionary stability of $HGT^+$ and $HGT^-$ populations

To study whether HGT is an evolvable trait, we will consider 1) if HGT can evolve de novo, and 2) if HGT can be evolutionarily maintained. For this, we extended the two-variable model of one species to a four-variable model of two species: a HGT$^+$-species that engages in HGT, and a HGT$^-$-species that does not (*Figure 1B*, *Equation 25-28*). We analysed under what conditions the HGT$^+$-species can invade an equilibrium of the HGT$^-$-species, and vice versa. We found that HGT can only evolve for an enrichable gene, but is evolutionarily maintained for both enrichable and rescuable genes. The following paragraphs will elaborate on how these results are derived:

Consider a HGT$^+$-species ($C^+$, $N^+$) and a HGT$^-$-species ($C^-$, $N^-$) that differ in their HGT rate $h$, but are identical otherwise. The dynamics of the density of cells carrying and not carrying the gene of the two species can be described by the following equations:

$$\frac{dC^-}{dt} = (1+b)C^- - lC^- - \phi C^- \tag{25}$$

$$\frac{dN^-}{dt} = N^- + lC^- - \phi N^- \tag{26}$$

$$\frac{dC^+}{dt} = (1+b-ch)C^+ - lC^+ + hN^+(C^- + C^+) - \phi C^+ \tag{27}$$

$$\frac{dN^+}{dt} = (1-ch)N^+ + lC^+ - hN^+(C^- + C^+) - \phi N^+ \tag{28}$$

$$\phi = (1+b)C^- + N^- + (1+b-ch)C^+ + (1-ch)N^+ \tag{29}$$

$$C^- + N^- + C^+ + N^+ = 1 \tag{30}$$

Note that we include horizontal gene transfer from HGT$^-$-cells carrying the gene to HGT$^+$-cells that do not yet carry the gene. In other words, we consider a situation in which the propensity of HGT is determined by the recipient cell, and not by the donor. This is inspired by for instance the process of transformation, in which the recipient cell 'decides' whether or not it takes up extracellular DNA.

If HGT is evolvable de novo, the HGT$^+$ species should be able to invade a HGT$^-$ population in steady state. In other words, the equilibrium state $(C^-, N^-, C^+, N^+) = (\widehat{C}^-, \widehat{N}^-, 0, 0)$ should be unstable.

Around the equilibrium $(\widehat{C}^-, \widehat{N}^-, 0, 0)$, the dynamics of the HGT$^+$-species are linearly approximated by

$$\begin{pmatrix} \frac{dC^+}{dt} \\ \frac{dN^+}{dt} \end{pmatrix} \approx \mathbf{J} \begin{pmatrix} C^+ \\ N^+ \end{pmatrix},$$

where

$$\mathbf{J} = \begin{pmatrix} 1+b-ch-l-\widehat{\phi} & h\widehat{C}^- \\ l & 1-ch-h\widehat{C}^- - \widehat{\phi} \end{pmatrix}.$$

The HGT$^+$-species can invade if the dominant eigenvalue of $\mathbf{J}$ is positive.

Note that the equilibrium densities of $\widehat{C}^-$ and $\widehat{N}^-$ depend on $b$ and $l$. As derived in the previous section,

$$if\, b \leq l, \quad \widehat{C}^- = 0 \quad and \quad \widehat{N}^- = 1, \quad while \tag{31}$$

$$if\, b > l, \quad \widehat{C}^- = 1 - \frac{l}{b} \quad and \quad \widehat{N}^- = \frac{l}{b}. \tag{32}$$

We will consider both possibilities separately.

In the case of unrescuable and rescuable genes ($0 < b \leq l$), the equilibrium densities of $\widehat{C}^-$ and $\widehat{N}^-$ are given by *Equation 31*. Then, $\widehat{\phi} = 1$ and the Jacobian matrix

$$\mathbf{J} = \begin{pmatrix} b-ch-l & 0 \\ l & -ch \end{pmatrix}.$$

The eigenvalues of $\mathbf{J}$ are $\lambda_1 = b - ch - l$ and $\lambda_2 = -ch$. The second eigenvalue $\lambda_2 < 0$ as long as HGT comes at some cost $c > 0$ (the HGT-rate $h$ of a HGT$^+$-species is always positive). At the same time, $\lambda_1$ is also negative because we consider genes with a small benefit, $0 < b \leq l$. Hence, we conclude that for unrescuable and more importantly for rescuable genes, an HGT$^+$-species cannot invade on a HGT$^-$-population at equilibrium, and HGT can hence never evolve de novo.

In the case of enrichable and indispensable genes ($b > l$), the equilibrium densities of $\widehat{C}^-$ and $\widehat{N}^-$ are given by *Equation 32*. Now, $\widehat{\phi} = (1+b)(1-\frac{l}{b}) + \frac{l}{b} = 1 + b - l$, and the Jacobian matrix

$$\mathbf{J} = \begin{pmatrix} -ch & h(1-\frac{l}{b}) \\ l & l-b-ch-h(1-\frac{l}{b}) \end{pmatrix}.$$

The eigenvalues of $\mathbf{J}$ should now be solved from

$$(-ch-\lambda)(l-b-ch-h(1-\frac{l}{b})-\lambda)-lh(1-\frac{l}{b})=0 \tag{33}$$

$$\Longleftrightarrow \lambda^2 - \lambda(l-b-2ch-h(1-\frac{l}{b})) + (bch-lch+c^2h^2+ch^2(1-\frac{l}{b})-lh(1-\frac{l}{b}))=0 \tag{34}$$

Let

$$\beta = l-b-2ch-h(1-\frac{l}{b}), \quad and \tag{35}$$

$$\gamma = bch-lch+c^2h^2+ch^2(1-\frac{l}{b})-lh(1-\frac{l}{b}). \tag{36}$$

Then, the eigenvalues of $\mathbf{J}$ are equal to $\lambda_{1,2} = \frac{1}{2}(\beta \pm \sqrt{\beta^2-4\gamma})$. Remember that we are interested in the sign of the dominant eigenvalue. If the eigenvalues are complex ($\beta^2 < 4\gamma$), the real part of the eigenvalues $\mathrm{Re}(\lambda_{1,2}) > 0$ if $\beta > 0$. If the eigenvalues are real, the dominant eigenvalue is $\lambda_1 = \frac{1}{2}(\beta + \sqrt{\beta^2-4\gamma})$, and $\lambda_1 > 0$ if $\beta > 0$ or $\sqrt{\beta^2-4\gamma} > \beta \Longleftrightarrow \gamma < 0$.

First, consider the possibility $\beta > 0$. Then we should have

$$l-b-2ch-h(1-\frac{l}{b})>0 \tag{37}$$

$$\Longleftrightarrow l-b > h(2c+(1-\frac{l}{b})). \tag{38}$$

This is however a contradiction, since we here deal with genes for which $b>l$ and hence $l-b<0$, but $\widehat{C}^- = 1-\frac{l}{b}>0$, $c>0$ and $h>0$. Hence, $\beta$ is always negative and the dominant eigenvalue is positive only if $\gamma<0$. From $\gamma<0$, we find

$$bch - lch + c^2h^2 + ch^2(1-\frac{l}{b}) - lh(1-\frac{l}{b})<0 \tag{39}$$

$$\Longleftrightarrow c(b-l+ch) + (ch-l)(1-\frac{l}{b})<0 \tag{40}$$

Trying to solve **Equation 40** for any value of $h$ would yield a complicated condition on the value of $b$. However, we can further simplify **Equation 40** by asking if a HGT$^+$-species with a very small (but positive) HGT-rate could invade. For $h = \epsilon \approx 0$, **Equation 40** reduces to

$$c(b-l) - l(1-\frac{l}{b})<0, \tag{41}$$

from which we can solve

$$c(b-l) - l(1-\frac{l}{b})<0 \tag{42}$$

$$\Longleftrightarrow cb^2 - l(c+1)b + l^2 <0 \tag{43}$$

$$\Longleftrightarrow (cb-l)(b-l)<0. \tag{44}$$

Since we consider enrichable and indispensable genes, with $b>l$, condition 44 can only be true if $cb<l \Longleftrightarrow b<l/c$, which is exactly the condition that separates enrichable from indispensable genes. Hence, we conclude that for enrichable genes ($l<b<l/c$), a HGT$^+$-species with a small but positive HGT-rate can always invade on a HGT$^-$-population at equilibrium, and that HGT can hence evolve de novo.

So far, we have determined under what conditions a HGT$^-$-population is evolutionarily stable. We can however ask the same for a HGT$^+$-population. In other words, even though it may not be reached by gradual evolution, can HGT be *maintained*? To answer this question, we next consider the evolutionary stability of the HGT$^+$-equilibrium: $(C^-, N^-, C^+, N^+) = (0, 0, \tilde{C}^+, \tilde{N}^+)$.

Again, the densities of $C^+$- and $N^+$-cells at equilibrium depend on the values of $b, l$ and $h$ (see **Equations 9–10** in the previous section):

$$\text{if } b \leq l-h, \quad \tilde{C}^+ =0 \ \text{ and } \ \tilde{N}^+ =1, \quad \text{while} \tag{45}$$

$$\text{if } b > l-h, \quad \tilde{C}^+ =1-\frac{l}{b+h} \ \text{ and } \ \tilde{N}^+ =\frac{l}{b+h}. \tag{46}$$

If $b \leq l-h$, the gene does not persist in the population and HGT hence does not confer any benefit, while still imposing a cost on the $N^+$-cells. Under these conditions, the $N^-$-cells, that do not carry the cost of HGT, will always be able to invade.

For the more interesting case in which the gene does persist in a HGT$^+$-population (**Equation 46**), we now linearise the dynamics of the HGT$^-$-species around the equilibrium:

$$\begin{pmatrix} \frac{dC^-}{dt} \\ \frac{dN^-}{dt} \end{pmatrix} = \mathbf{J}\begin{pmatrix} C^- \\ N^- \end{pmatrix}$$

$$\text{with } \mathbf{J} = \begin{pmatrix} (1+b)-l-\tilde{\phi} & 0 \\ l & 1-\tilde{\phi} \end{pmatrix}$$

$$and\ \tilde{\phi} = (1 + b - ch)(1 - \frac{l}{b+h}) + (1 - ch)\frac{l}{b+h} = (1 - ch) + b(1 - \frac{l}{b+h}).$$

Again, the HGT$^-$-species can invade if the dominant eigenvalue of **J** is positive, and hence the HGT$^+$-species of equilibrium is evolutionarily stable if both eigenvalues are negative. The eigenvalues of **J** are $\lambda_1 = 1 + b - l - \tilde{\phi}$ and $\lambda_2 = 1 - \tilde{\phi}$.

For the first eigenvalue, we find

$$\lambda_1 < 0 \tag{47}$$

$$\Longleftrightarrow 0 > 1 + b - l - \tilde{\phi} \tag{48}$$

$$\Longleftrightarrow 0 > 1 + b - l - (1 - ch) - b(1 - \frac{l}{b+h}) \tag{49}$$

$$\Longleftrightarrow 0 > bl + ch(b+h) - l(b+h) \tag{50}$$

$$\Longleftrightarrow lh > ch(b+h) \tag{51}$$

$$\Longleftrightarrow l > c(b+h) \tag{52}$$

$$\Longleftrightarrow c < \frac{l}{b+h} \tag{53}$$

Hence, this first eigenvalue is negative as long as the costs of HGT are not too large.

For the second eigenvalue, we find:

$$\lambda_2 < 0 \tag{54}$$

$$\Longleftrightarrow 0 > 1 - \tilde{\phi} \tag{55}$$

$$\Longleftrightarrow 0 > 1 - (1 - ch) - b(1 - \frac{l}{b+h}) \tag{56}$$

$$\Longleftrightarrow 0 > ch - b(1 - \frac{l}{b+h}) \tag{57}$$

$$\Longleftrightarrow ch < b(1 - \frac{l}{b+h}) \tag{58}$$

$$\Longleftrightarrow c < \frac{b(1 - \frac{l}{b+h})}{h}. \tag{59}$$

Remember that we considered a HGT$^+$-population in which the gene can persist, that is $b + h > l$. Hence $\frac{l}{b+h} < 1$ and the right hand side in **Equation 59** is positive. Hence, we can again conclude that there are some non-zero costs for which $\lambda_2$ is negative.

Combining the results in **Equations 53 and 59**, we see that for some costs, HGT can be maintained. For rescuable genes with costs that satisfy conditions **Equations 53 and 59**, there is an Allee effect with respect to HGT: HGT can be evolutionarily maintained, but it cannot evolve de novo. This result can be intuitively understood. Small (invading) HGT$^+$-populations pay the continuous costs for HGT, but hardly ever interact with their conspecifics, and hence the positive fitness effects of maintaining the slightly beneficial gene are too small to overcome the costs for HGT. Higher fitness can only be achieved when the population size is

large enough, such that the benefits conferred by HGT outweigh its costs. The presence of an Allee effect was confirmed by numerically integrating *Equations 25-28* for different initial conditions. We then indeed see that the system converges to different equilibria depending on the initial frequency of HGT$^+$-cells (see *Figure 3*).

## Part II: Supplementary results and figures

### Similar gene classes exist with different assumptions for the cost of HGT

In the main text, we assume that the costs for HGT scale proportional with the rate of HGT. Moreover, we assume that the costs are always present, that is also when no carriers are present to interact with. The assumption is that both carriers and non-carriers can take up DNA, but do so to no avail when no carrier DNA is present. Below, we show what the effect is on population growth rates when these assumptions are varied. Specifically, we consider a scenario with constant costs, and a scenario where the costs are only payed when carriers are present from which to take up DNA.

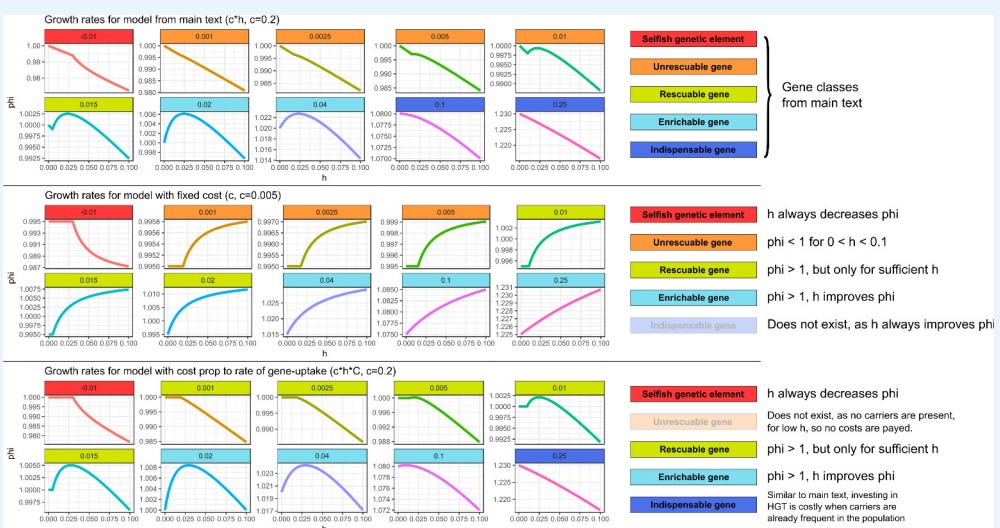

**Appendix 1—figure 1.** Similar gene classes exist with different assumptions for the cost of HGT In the top panel (with costs as in the main text), we can observe all the gene classes discussed in the main text. In the middle panel (with fixed costs), HGT always improves the growth rates for genes with $b>0$. However, for low $b$, the growth rate is lower than one even for very high $h$. Therefore, it is not adaptive to invest in costly HGT (the maximal growth rate is 1.0 with $h = 0$). Thus, for this range of $h$'s, we observe a similar 'unrescuable' gene class as the one discussed in the main text. When growth rates *can* be improved within this range of $h$'s, the genes can be compared to the rescuable genes or enrichable genes, depending on whether or not HGT is required for the gene to persist. Indispensable genes do not exist under this regime, as HGT always increases growth rates when $b>0$. We argue, however, that a constant cost with extremely high rates of HGT is fairly unrealistic, as every HGT-event carries inherent risks, for example by chromosome disruptions, cytotoxicity, and the integration of SGEs. In the bottom panel, we investigate a scenario where costs are only present when actually taking up DNA from carrier cells (*i.e.* instead of c*h, the costs are c*h*C). Under this scenario, most gene classes from the main text exist. Unrescuable genes however do not exist, as there are no cost if no carriers are present. Similarly, the Allee effect as discussed in the main text is also not present, as this Allee effect requires costs in the absence of donor cells.

## Comparing our results with conjugation-like scenarios

In the main text, we discussed how HGT may not be able to evolve for rescuable genes due to the Allee effect: HGT is only adaptive if the gene is present, but the gene can only be present with HGT. Here, we modified our model to represent a scenario more similar to bacterial conjugation, by assuming that $C^-$ (i.e. a carrier without the ability for HGT) cannot act as a donor. Interestingly, under these assumptions we found that the enrichable gene class also shows the Allee effect (see image below). This is intuitive, as in this scenario the $HGT^+$ strains lack donors to interact with, regardless of the gene already being maintained by the resident $HGT^-$ strain. Thus, similar to the results discussed in the main text, $HGT^+$ cannot invade. Indeed, we found that this conjugation-like scenario has different dynamics for the evolution of HGT, where both rescuable and enrichable genes require the 'nucleation events' presented in the main text.

Allee effect only present for rescuable gene when C⁻ (a carrier that cannot engage in HGT) can act as a donor (*e.g.* **transformation**)

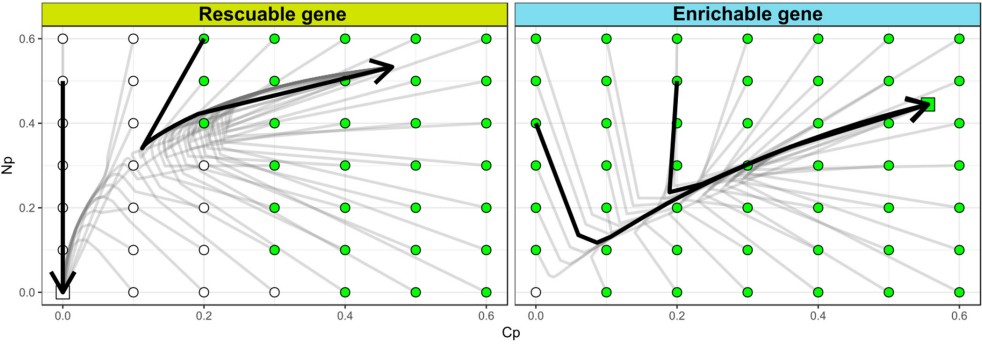

Allee effect also present for enrichable genes when C⁻ (a carrier that cannot engage in HGT) cannot act as a donor (*e.g.* **conjugation**)

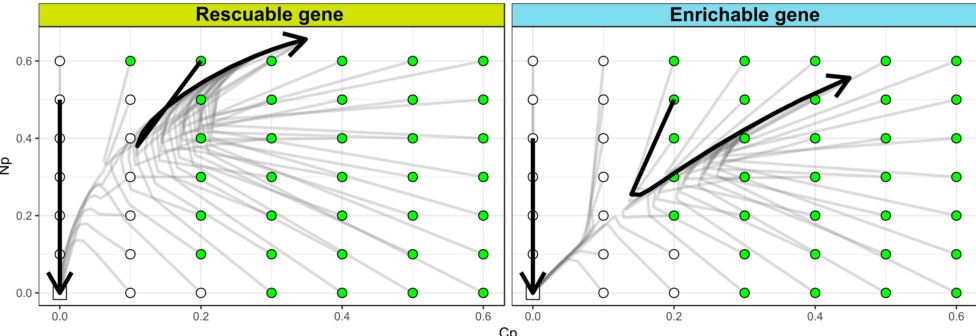

**Appendix 1—figure 2.** The Allee effect for de novo evolution of HGT is also present for enrichable genes when $C^-$ (a carrier that cannot engage in HGT) cannot act as a donor.

Interestingly, when studying the emergence of a 'gene-sharing' community (using a slow rate of gene-influx as in the main text), we found that it most readily evolved when HGT happened at intermediate physical distances. After 50.000 time steps, both the conjugation-like (C- cannot act as donor) and the transformation-like (C- can act as donor) mechaninisms of HGT show that only a subset of the populations managed to evolve HGT for low physical distances(see figure below). The details of this figure of course depend on $b$, as slightly more beneficial genes may spread easier. It is however interesting to consider how the 'gene-sharing' discussed in the main text, works best when the genes can be picked up from afar, ensuring that the clumping of carrier cells (see *Figure 5*) is alleviated. This distance-effect is of course not applicable to conjugation, as physical contact is required between cells in order to

conjugate. Finally, note that other than these change in the evolutionary accessibility of HGT, the other results discussed in the main text remain qualitatively similar.

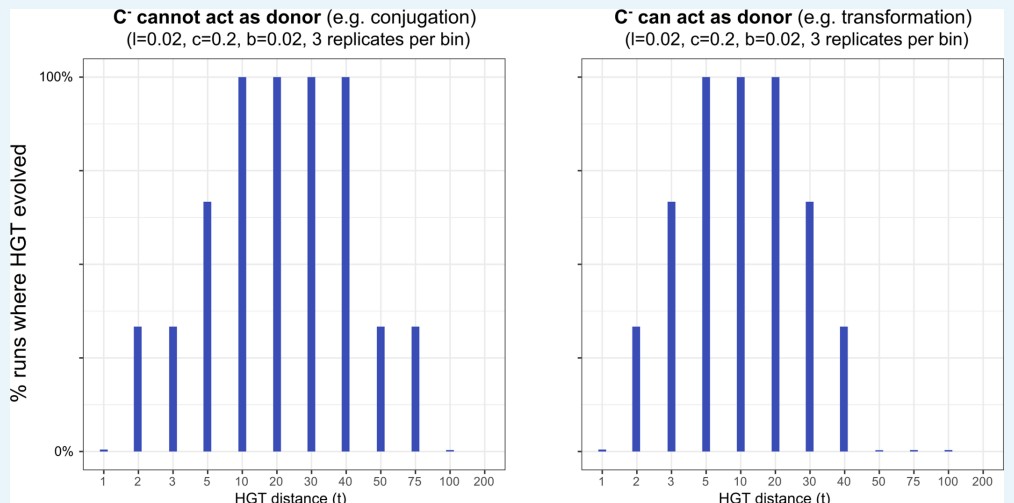

**Appendix 1—figure 3.** HGT evolves most readily when transfer happens between donor/recipient pairs of intermediate distances. On the left-hand side, we consider this for a conjugation-like scenario ($C^-$, a carrier that cannot engage in HGT, cannot act as a donor), while on the right-hand side we consider this for a transformation-like scenario ($C^-$, a carrier that cannot engage in HGT, *can* act as a donor).

## In the well-mixed IBM, HGT only evolves for enrichable genes

In the main text we discussed that HGT cannot evolve for genes that cannot persist without HGT. For these genes, a lack of donor cells does not allow mutants that engage in HGT to get a significant fitness benefit, even when they actually *do* carry the beneficial gene. To get over this so-called Allee effect, a large number of gene-carrying individuals has to simultaneously start engaging in HGT. We have also shown that, in the spatially structure populatfions, HGT *does* evolve for genes that could not persist without HGT, as it is more likely that the lack of donor cells is, at least locally, overcome. This supplementary figure summarises this result, by showing that, even though HGT does evolve for enrichable genes under well-mixed conditions, it indeed fails to evolve for rescuable genes. As discussed in the previous paragraph, a conjugation-like scenario causes the Allee effect to also exist for enrichable genes. Indeed, we observed that well-mixed populations could not evolve HGT for enrichable genes (i.o.w. HGT could not evolve for any gene category!), but spatially structured populations would readily overcome this by emergent gene-sharing communities.

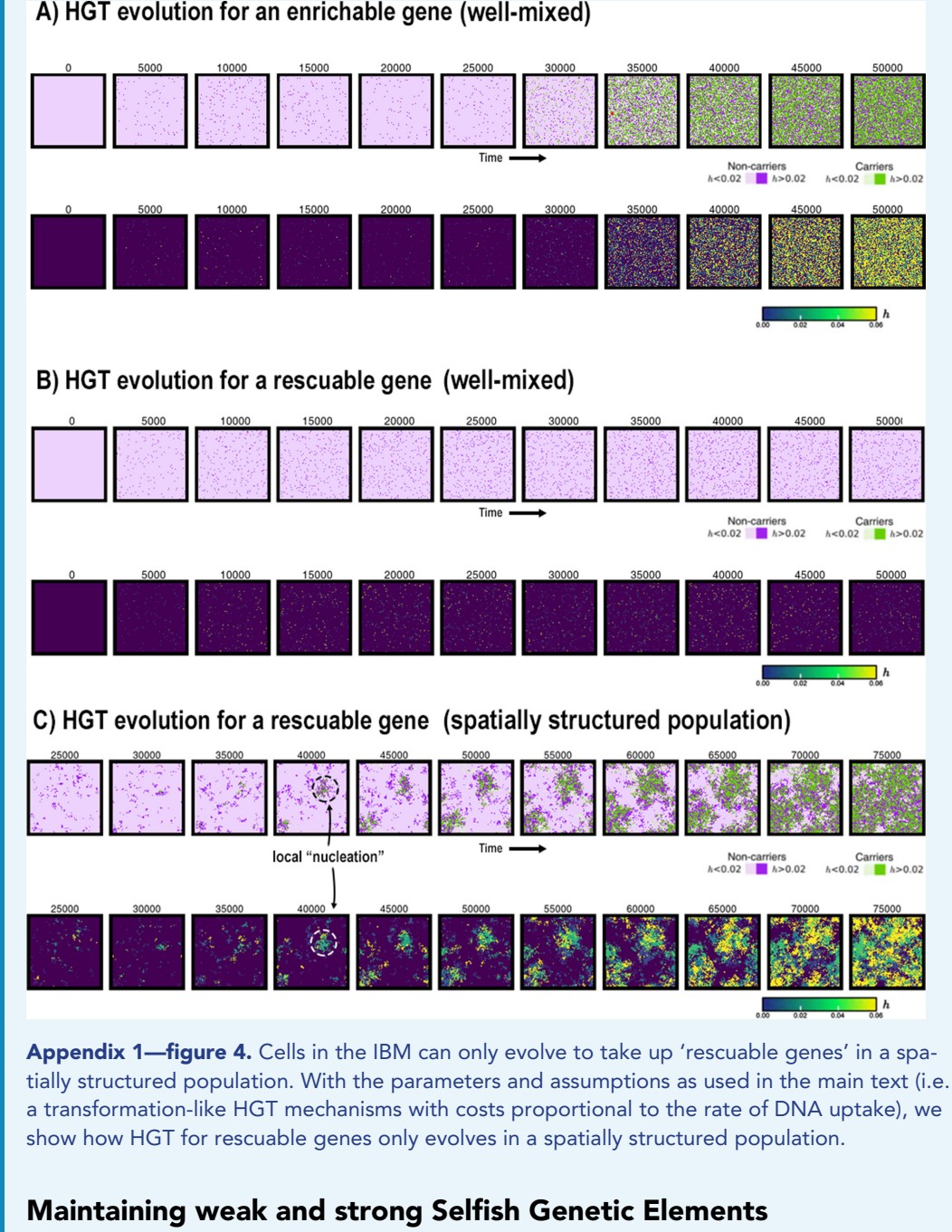

**Appendix 1—figure 4.** Cells in the IBM can only evolve to take up 'rescuable genes' in a spatially structured population. With the parameters and assumptions as used in the main text (i.e. a transformation-like HGT mechanisms with costs proportional to the rate of DNA uptake), we show how HGT for rescuable genes only evolves in a spatially structured population.

## Maintaining weak and strong Selfish Genetic Elements

In the main text we have discussed how SGEs can coexist along-side their hosts and slightly beneficial genes, even when the their fitness-penalty is greater than the benefit of the gene. However, this was only observed in the spatially structured model, as illustrated in the figure below.

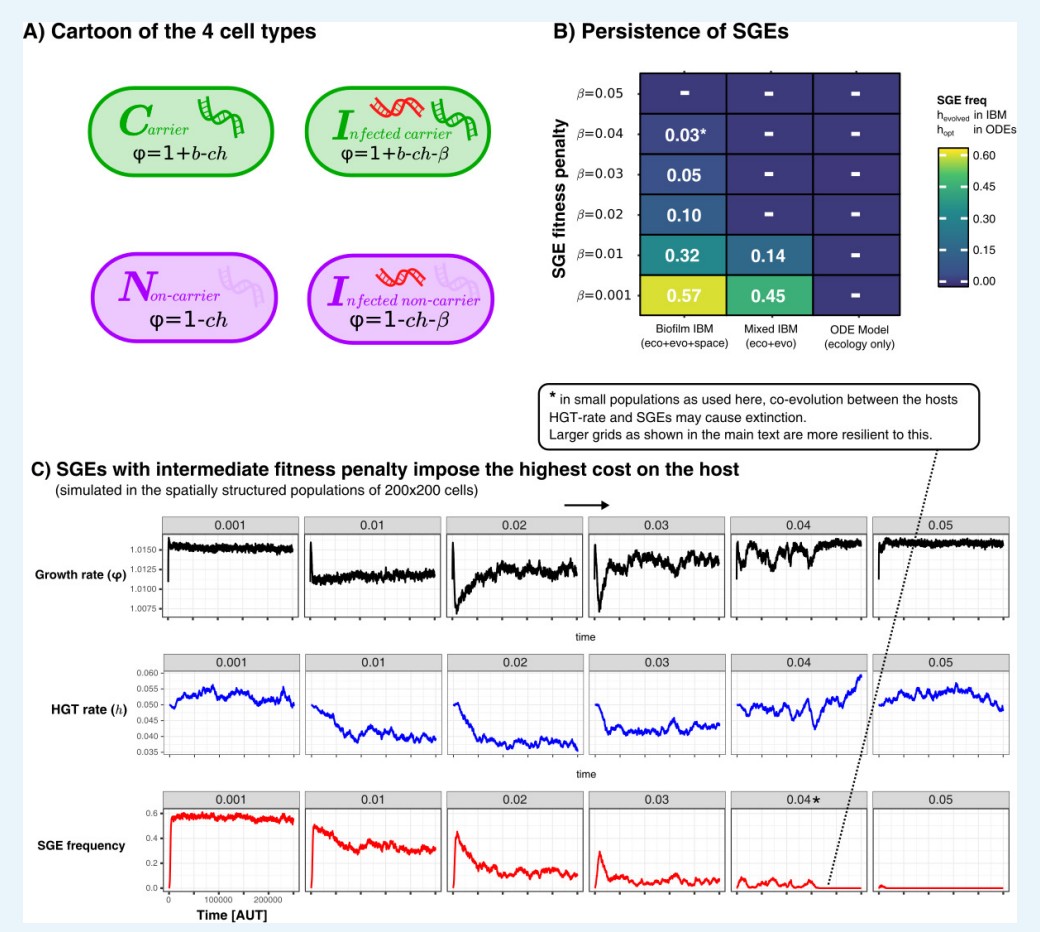

**Appendix 1—figure 5.** Persistence of SGEs in various implementations of our model. A shows a cartoon of the cell types, between which competition was modelled in a various ways. In B we show for these different implementations how many SGEs persist within the populations for SGEs with different penalties. For the IBM, we simulated for 250.000 time steps and calculated the average SGE-frequency in the final 100 generations. For the ODE model, we chose the optimal rate of HGT ($h_{opt}$), and numerically integrated the equilibrium concentrations of infected cells. Finally, C shows the temporal dynamics of the growth rate ($\phi$), HGT-rate ($h$), and the SGE frequency, in the spatially structured simulations. As this parameter sweep had slighly smaller populations sizes as used in the main text, the strong SGE could eventually go extinct (this is annotated with an asterisk).

Equations for *Figure 5A*

$$
\begin{aligned}
\frac{\mathrm{d}C}{\mathrm{d}t} &= \gamma(1+b-ch)C - lC + lD + h(NC + 0.5ND - CP - CD) - \phi C \\
\frac{\mathrm{d}N}{\mathrm{d}t} &= \gamma(1-ch)N + lC + lP - hN(C+P+D) - \phi N \\
\frac{\mathrm{d}P}{\mathrm{d}t} &= \gamma(1-\beta-ch)P + lD - lP + h(NP + ND/2 - CP - PD) - \phi P \qquad (60) \\
\frac{\mathrm{d}D}{\mathrm{d}t} &= \gamma(1+b-\beta-ch)D - lD2 + h(CP + CD + PC + PD) - \phi D \\
\phi &= \gamma((1+b-ch)C + (1-ch)N + (1-\beta-ch)P + (1+b-\beta-ch)D)
\end{aligned}
$$

# Strong SGEs fail to spread/persist in the population at low HGT-distances

In the main text we have discussed how we found that strong SGEs (genetic parasites with a greater penalty than the beneficial gene) could nevertheless stably coexist with an evolving population of cells. However, this persistence of SGEs relies on their ability to escape to new susceptible hosts who have not experienced SGEs for some time (and therefore have evolved

elevated HGT rates). In this supplementary figure, it is indeed seen how the distance influences the spread/persistence of SGEs. If the distance between donor and recipient is very local (d=1), SGEs cannot spread even while they are still fluxing in (top row). For an intermediate HGT-distance (1<$d$<10), the SGEs persist for a bit as long as they flux in, but die out when influx is stopped (middle row). For larger HGT distances ($d$>10), we found that SGEs can persist even after the influx was stopped.

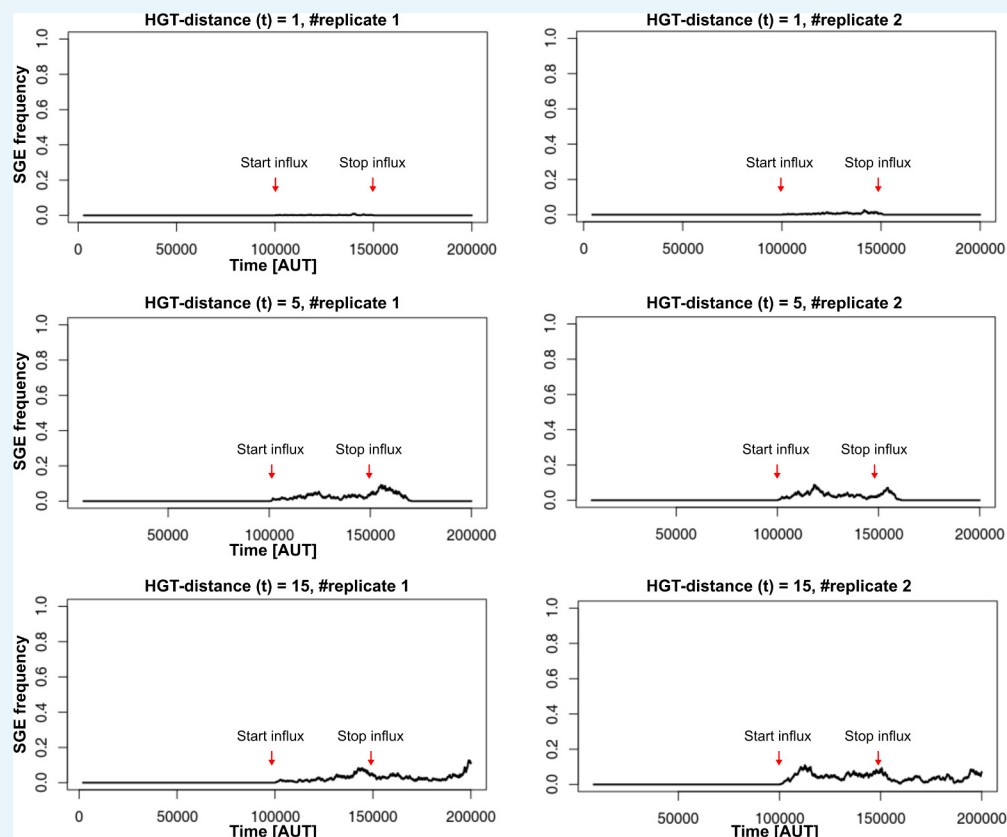

**Appendix 1—figure 6.** SGEs persistence can depend on the distance of HGT and on the continued influx of SGEs.(parameters used as in main text).

van Dijk *et al*. eLife 2020;9:e56801. DOI: https://doi.org/10.7554/eLife.56801

**Appendix 1—table 1.** Values of $h_{opt}$ that are attained when testing if HGT could be maintained. Starting from $h = 0.05$, we ran the model with evolving $h$ to test if HGT would be maintained. Positions in this table correspond to **Figure 4A** in the main text. If $h_{evolved}$ persisted at values > 0.001 after 500.000 time steps, we call HGT maintainable. Otherwise, the value in this table is 0.

| (b) | IBM (well-mixed) | IBM (d = 10) | IBM (d = 1) | IBM (d = 0) |
|---|---|---|---|---|
| 0.2025 | 0 | 0 | 0 | 0 |
| 0.1975 | 0 | 0 | 0 | 0.0035 |
| 0.1925 | 0 | 0 | 0 | 0.009 |
| 0.1875 | 0 | 0 | 0 | 0.016 |
| 0.1825 | 0 | 0 | 0 | 0.017 |
| 0.1775 | 0 | 0 | 0 | 0.022 |
| 0.1725 | 0 | 0 | 0 | 0.024 |
| 0.1675 | 0 | 0 | 0 | 0.028 |
| 0.1625 | 0 | 0 | 0.003 | 0.033 |
| 0.1575 | 0 | 0 | 0.003 | 0.033 |
| 0.1525 | 0 | 0 | 0.0035 | 0.035 |
| 0.1475 | 0 | 0 | 0.004 | 0.035 |
| 0.1425 | 0 | 0 | 0.004 | 0.035 |
| 0.1375 | 0 | 0 | 0.006 | 0.035 |
| 0.1325 | 0 | 0 | 0.008 | 0.035 |
| 0.1275 | 0 | 0 | 0.01 | 0.036 |
| 0.1225 | 0 | 0 | 0.011 | 0.036 |
| 0.1175 | 0 | 0 | 0.014 | 0.036 |
| 0.1125 | 0 | 0 | 0.016 | 0.036 |
| 0.1075 | 0 | 0 | 0.018 | 0.036 |
| 0.1025 | 0 | 0.002 | 0.02 | 0.036 |
| 0.0975 | 0.002 | 0.004 | 0.021 | 0.036 |
| 0.0925 | 0.004 | 0.006 | 0.022 | 0.037 |
| 0.0875 | 0.006 | 0.006 | 0.023 | 0.037 |
| 0.0825 | 0.008 | 0.008 | 0.023 | 0.037 |
| 0.0775 | 0.01 | 0.01 | 0.025 | 0.037 |
| 0.0725 | 0.012 | 0.012 | 0.026 | 0.037 |
| 0.0675 | 0.014 | 0.014 | 0.027 | 0.038 |
| 0.0625 | 0.016 | 0.017 | 0.028 | 0.038 |
| 0.0575 | 0.018 | 0.018 | 0.028 | 0.038 |
| 0.0525 | 0.02 | 0.021 | 0.028 | 0.038 |
| 0.0475 | 0.021 | 0.021 | 0.029 | 0.038 |
| 0.0425 | 0.023 | 0.023 | 0.029 | 0.039 |
| 0.0375 | 0.024 | 0.025 | 0.03 | 0.041 |
| 0.0325 | 0.025 | 0.025 | 0.03 | 0.041 |
| 0.0275 | 0.025 | 0.026 | 0.031 | 0.041 |
| 0.0225 | 0.025 | 0.025 | 0.032 | 0.041 |
| 0.0175 | 0.024 | 0.025 | 0.029 | 0.037 |
| 0.0125 | 0.023 | 0.024 | 0 | 0 |

*Appendix 1—table 1 continued on next page*

*Appendix 1—table 1 continued*

| (b) | IBM (well-mixed) | IBM (d = 10) | IBM (d = 1) | IBM (d = 0) |
|---|---|---|---|---|
| 0.0075 | 0 | 0 | 0 | 0 |
| 0.0025 | 0 | 0 | 0 | 0 |
| −0.0025 | 0 | 0 | 0 | 0 |
| −0.0075 | 0 | 0 | 0 | 0 |
| −0.0125 | 0 | 0 | 0 | 0 |

