## [Decision Letter]

**Acceptance summary:**

This paper presents a computational model of horizontal gene transfer (HGT) in bacteria that explores the parameter space that would allow it to evolve. Interestingly, their model shows that even when costly, HGT is most like to evolve for genes with slight benefits to their hosts, which would be lost in the absence of HGT. HGT for these genes required a spatial context to evolve, but once these conditions were fulfilled, it evolved even in the presence of selfish genetic elements, which carry no benefit to their hosts.

**Decision letter after peer review:**

Thank you for submitting your article "Slightly beneficial genes are retained by collectives evolving Horizontal Gene Transfer despite selfish elements" for consideration by *eLife*. Your article has been reviewed by three peer reviewers, one of whom is a member of our Board of Reviewing Editors, and the evaluation has been overseen by Patricia Wittkopp as the Senior Editor. The following individuals involved in review of your submission have agreed to reveal their identity: Paul G Higgs (Reviewer #2).

The reviewers have discussed the reviews with one another and the Reviewing Editor has drafted this decision to help you prepare a revised submission.

Summary:

This study uses a combination of two modelling approaches (differential equation models and IBMs) to ask how HGT can evolve de novo, and what kind of genes will allow for selection and maintenance of HGT. The analysis reveals that depending on the benefit (or harm) they provide, genes can be categorised into indispensable, enrichable, rescuable, unrescuable or selfish. They analyse these different gene types and show that HGT is more likely to evolve and be maintained for "enrichable" genes that have a small benefit to their carriers or for both enrichable and rescuable genes if a spatial version of the model is considered.

Essential revisions:

All reviewers agreed that this is a very interesting paper that gives some useful ways to think about HGT (the gene categories) and important insights into its evolution. We also agreed that the modelling approaches used were interesting, informative and appropriate. The paper is well-written, in particular the figures, which are very visually appealing.

A first key point is clarifying what mechanism of HGT the model is capturing: how is transfer occurring? What does the cost represent and how are genes acquired or lost? The paper contains mixed messages as to whether HGT occurs through plasmids or by transformation. According to the definition of "additive HGT", it should be through plasmids, and not by transformation, which requires recombination (Introduction paragraph two). But then other parts of the model (like acquiring genes from cells a certain distance away) suggest transformation. This choice will be important in determining the model structure, the rate of loss of the genetic element and its costs and benefits as detailed below.

Regarding the cost, what does it represent, and why is the cost proportional to the rate of transfer (ch) rather than just a constant cost (c)? If the HGT mechanism is conjugation, there is a cost of replicating the plasmid inside the cell, but the cost is constant – it doesn't depend on how fast the genes are transferred, just on how many plasmids are in the cell. Alternatively, if fragments of DNA are taken up from the environment, maybe there is a small cost to expressing uptake genes, and this might be proportional to h, although one would expect the main fitness benefits and costs are associated with the genes/junk/SGEs that are acquired by HGT rather than simply from having the ability to acquire genes by HGT.

Regarding gene loss, how it is modelled will also depend on the mechanism of HGT. If the gene is on a chromosome, then loss could be deletion of a gene during genome replication, or deleterious mutation that stops the function of the gene. On the other hand, if the beneficial gene is on a plasmid, loss of the beneficial gene is loss of the plasmid. This will clearly also affect the magnitude of the parameter value. In Equation 3, the loss rate from C^-^ is the same as the loss rate from C^+^, which would be true if the gene is on a chromosome. But if that is the case, the gene that allows HGT should also be on the chromosome (contradicting with the additive HGT assumption) and it should also be lost sometimes, in which case a C^+^ could change to a C^-^, or an N^+^ to an N^-^. Another way to say this is "where does the C^-^ come from?" If the beneficial gene were gained by HGT then it must have been gained by a + cell, then the C^-^ was created by deletion of the gene that previously allowed HGT.

Another way to view the model is that the C^+^ represents a cell where the beneficial gene is on a plasmid that can be transferred by conjugation, whereas the C^-^ represents a cell where the beneficial gene is already on a chromosome and cannot be transferred. In that case the loss rate from the chromosome by deletion should be different from the loss by losing the plasmid. Losing the plasmid means losing the benefit and the cost and the ability to transfer all at the same time. There is an interesting paper of Bergstrom that discusses the conditions for maintenance of a plasmid with a beneficial gene (Bergstrom et al., 2000). It would be useful to discuss similarities and differences of the present model with the Bergstrom model. An important part of the Bergstrom model is the transfer of the beneficial gene to the chromosome. This seems to be analogous to the mutation of a C^+^ to a C^-^ in this model. In Equation 3 the rate of gain of the beneficial gene is hN^+^(C^-^ + C^+^). This means that C^-^ cells can donate the gene. This would apply if the gene is acquired by transformation, rather than conjugation. How would this change in the conjugation case? The rate of gain would be hN+C^+^ only. The N^+^ cannot gain from the C^-^. Does that mean the plasmid cannot spread? I think the assumption that the cost of HGT is proportional to h is relevant here. If the cost were a constant c (not ch) then a plasmid with high enough h would invade whatever the cost to the cells.

Finally on loss, the current choice of parameter for the rate of loss is odd. The authors claim that their results hold for more realistic values (subsection “Parameters used”) but making the parameter-region where HGT is adaptive for the host cells narrower. Is the parameter region large enough to be biologically relevant?

A second point that caused confusion is that the model assumes that each gene will either have a beneficial or a deleterious effect all the time. In reality, of course, MGEs lie on a parasitism-mutualism continuum relationship with the host bacterium because the “beneficial” gene will not be beneficial all the time (see Harrison et al., 2015, Discussion paragraph two). The same MGE transferring the beneficial gene may be an SGE depending on the bacterial host or the environment. The model thus considers a rather artificial case in which a single gene always has the same effect on its host. We see that it is a valid simplification to simulate each gene benefit individually, understand the conditions in which it may spread, and thereby infer that as long as genes with such benefits exist, then HGT should evolve. But this last logical step is not spelled out clearly enough in the paper.

Whether or not the gene is beneficial may also change the mechanism of transfer, where genes that are beneficial in the long term are more likely to integrate on the chromosome and reducing the rate of loss possibly by orders of magnitude. This last point is crucial, because if the rate of loss is low enough then slightly beneficial genes could be maintained in the absence of HGT. In other words, the authors may want to consider a case where the rate of loss differs depending on the costs and benefits of the gene in a given context.

A related point is how deleterious can SGEs become without disrupting HGT? This seems to be explored in Appendix 1—figure 5B and C. Is the result that SGEs would simply disappear if they are too deleterious? There seems to be a suggestion in the supplement that grid size may change this result. It would also be good to expand on this in the main text.

In sum, in order for the manuscript to be accepted in *eLife*, we ask the authors to at least do the following:

1) Include a detailed discussion at the beginning of the paper about the assumptions of the model and which biological situation is represented. Are you simulating HGT by conjugation or transformation? Who decides whether transfer is possible – the donor, the receiver, or both?

2) The authors should more explicitly explain where their parameter choices for costs, benefits, rates of loss, etc. and how they relate to the chosen mechanism of HGT. Also, why is it reasonable to assume that they are independent of each other?

3) Revise the choices of cost, benefit and rates of loss according to the biological situation and taking into account the ideas discussed above. Will the conclusions hold and will they be biologically relevant (if the parameter range is very small, is HGT likely to evolve)?

4) More fully cite the relevant literature on HGT costs and how they arise (TREE 2013 and San Millan and MacLean, Microbiology Spectrum 2017)

5) Flesh out what happens if SGEs are more deleterious.

---

## [Author Response]

Essential revisions:All reviewers agreed that this is a very interesting paper that gives some useful ways to think about HGT (the gene categories) and important insights into its evolution. We also agreed that the modelling approaches used were interesting, informative and appropriate. The paper is well-written, in particular the figures, which are very visually appealing.A first key point is clarifying what mechanism of HGT the model is capturing: how is transfer occurring? What does the cost represent and how are genes acquired or lost? The paper contains mixed messages as to whether HGT occurs through plasmids or by transformation. According to the definition of "additive HGT", it should be through plasmids, and not by transformation, which requires recombination (Introduction paragraph two). But then other parts of the model (like acquiring genes from cells a certain distance away) suggest transformation. This choice will be important in determining the model structure, the rate of loss of the genetic element and its costs and benefits as detailed below.

In our initial manuscript, we indeed are not clear on this. We modified the text to point out that the mechanism we study is most similar to HGT through bacterial transformation (Introduction paragraph three, Results paragraph one and Materials and methods first paragraph). However, we would also like to point out that many of the results in our manuscript may apply to HGT in the Abstract, generating useful search images for experimentalists and bioinformaticians. To show that our results could be insightful when different assumptions are considered, we added two supplementary paragraphs (and three figures) where discuss our model in light of different assumptions. We show that under various assumptions it is still fitting to talk about genes being rescuable (or not), indispensable (or not), and that the evolutionary “paradox” (*i.e.* HGT is needed for the gene to persist, but the gene only persist with HGT) applies even more when C^-^ cannot act as a donor, as as the case with conjugation.

Next, we now elaborate on how additive gene transfer can also occur through the uptake of DNA (*i.e.* through flanking homology / non-homologous recombination / transposons). This is relevant, as such additive incorporation of foreign DNA is essential for SGEs to be able to persist, which is why our model best applies to additive HGT (rather than replacing HGT).

Regarding the cost, what does it represent, and why is the cost proportional to the rate of transfer (ch) rather than just a constant cost (c)? If the HGT mechanism is conjugation, there is a cost of replicating the plasmid inside the cell, but the cost is constant – it doesn't depend on how fast the genes are transferred, just on how many plasmids are in the cell. Alternatively, if fragments of DNA are taken up from the environment, maybe there is a small cost to expressing uptake genes, and this might be proportional to h, although one would expect the main fitness benefits and costs are associated with the genes/junk/SGEs that are acquired by HGT rather than simply from having the ability to acquire genes by HGT.

We now discuss that the costs are those that are applicable for transformation / natural competence (Introduction paragraph two and three, Materials and methods paragraph one). Most notably, (i) the expression and operation of DNA uptake machinery is costly, and (ii) the physiological state of competence is shown to stop cell division or paralyse core metabolic processes (see for example Ephrussi-Taylor and Freed, 1964; Bergé et al., 2017, Villa et al., 2019). Moreover, we argue that every HGT-event carries a risk, for example by chromosome disruptions, cytotoxicity, and the integration of SGEs. The latter SGEs are of course explicitly discussed in the remainder of the manuscript.

We have also considered the model for constant costs, which is now included in a supplementary paragraph (and figure). These new supplementary results (Appendix 1 Part II) show that similar gene classes are observed for different assumptions for the costs of HGT. However, with fixed cost, the indispensable gene class is lost, as more and more HGT will always increase the frequency of carriers in the population. We consider these constant costs unrealistic however, as we expect that very high rates of HGT would eventually be a burden for the host cells (as explained above)

Regarding gene loss, how it is modelled will also depend on the mechanism of HGT. If the gene is on a chromosome, then loss could be deletion of a gene during genome replication, or deleterious mutation that stops the function of the gene. On the other hand, if the beneficial gene is on a plasmid, loss of the beneficial gene is loss of the plasmid. This will clearly also affect the magnitude of the parameter value. In Equation 3, the loss rate from C^-^ is the same as the loss rate from C^+^, which would be true if the gene is on a chromosome. But if that is the case, the gene that allows HGT should also be on the chromosome (contradicting with the additive HGT assumption) and it should also be lost sometimes, in which case a C^+^ could change to a C^-^, or an N^+^ to an N^-^. Another way to say this is "where does the C^-^ come from?" If the beneficial gene were gained by HGT then it must have been gained by a + cell, then the C^-^ was created by deletion of the gene that previously allowed HGT.

As mentioned above, it is our understanding that transformation can also lead to additive HGT.

That is an interesting point. Indeed, given that the benefit of rescuing slightly beneficial genes is, by definition, also small, the gene(s) responsible for HGT may require HGT too. We have considered this secondary effect earlier in the project, but have not yet extensively analysed its implications. Although we agree with the reviewers that this would be an interesting extension of our analysis, we feel it would cloud the message of our current manuscript.

Another way to view the model is that the C^+^ represents a cell where the beneficial gene is on a plasmid that can be transferred by conjugation, whereas the C^-^ represents a cell where the beneficial gene is already on a chromosome and cannot be transferred. In that case the loss rate from the chromosome by deletion should be different from the loss by losing the plasmid. Losing the plasmid means losing the benefit and the cost and the ability to transfer all at the same time. There is an interesting paper of Bergstrom that discusses the conditions for maintenance of a plasmid with a beneficial gene (Bergstrom et al., 2000). It would be useful to discuss similarities and differences of the present model with the Bergstrom model. An important part of the Bergstrom model is the transfer of the beneficial gene to the chromosome. This seems to be analogous to the mutation of a C+ to a C^-^ in this model. In Equation 3 the rate of gain of the beneficial gene is hN^+^(C^-^ + C^+^). This means that C^-^ cells can donate the gene. This would apply if the gene is acquired by transformation, rather than conjugation. How would this change in the conjugation case? The rate of gain would be hN+C^+^ only. The N^+^ cannot gain from the C^-^. Does that mean the plasmid cannot spread? I think the assumption that the cost of HGT is proportional to h is relevant here. If the cost were a constant c (not ch) then a plasmid with high enough h would invade whatever the cost to the cells.

Thank you for pointing that out. We added a paragraph to the Discussion to elaborate on the comparison between conjugation (i.e. Bergstrom) and transformation (i.e. our work), and therein also speculate on promising directions for future modelling, to better contrast the two mechanisms.

Changing whether or not C^-^ can act as a donor does not change our initial analysis, which was done in the simpler model from Figure 1A. However, it does change some of the implications for evolution of HGT. The Allee effect (as discussed in the main text), now also exists for the enrichable genes when C^-^ cannot act as a donor. An extra supplementary paragraph (and figure) now illustrates this. Interestingly, these new supplementary result suggest that the act of “rescuing” genes may be even more relevant with respect to conjugation.

Finally on loss, the current choice of parameter for the rate of loss is odd. The authors claim that their results hold for more realistic values (subsection “Parameters used”) but making the parameter-region where HGT is adaptive for the host cells narrower. Is the parameter region large enough to be biologically relevant?

The absolute value of the gene loss parameter indeed appears rather high, but bear in mind that the important point is the relationship between the costs and this loss rate. HGT is adaptive when *b*<*l*/*c*, meaning that our results may also apply to lower loss, depending on the costs. We do not know of any quantitative information to judge this issue on the balance between loss and costs accurately.

A second point that caused confusion is that the model assumes that each gene will either have a beneficial or a deleterious effect all the time. In reality, of course, MGEs lie on a parasitism-mutualism continuum relationship with the host bacterium because the “beneficial” gene will not be beneficial all the time (see Harrison et al., 2015, Discussion paragraph two). The same MGE transferring the beneficial gene may be an SGE depending on the bacterial host or the environment. The model thus considers a rather artificial case in which a single gene always has the same effect on its host. We see that it is a valid simplification to simulate each gene benefit individually, understand the conditions in which it may spread, and thereby infer that as long as genes with such benefits exist, then HGT should evolve. But this last logical step is not spelled out clearly enough in the paper.

Yes, that last logical step is important to point out. We elaborate on this in the (new) second paragraph of the Discussion. Here, we also discuss how much of our work was based on one of our earlier work. In this earlier eco-evolutionary model, we do study the co-evolution of beneficial genes and parasitic DNA, by allowing genes to evolve their own rate of transfer (van Dijk and Hogeweg, 2015). Here, we found that rarely beneficial toxin genes would evolve very high gene mobility, while their corresponding resistance factors evolved minimal gene mobility.

With the current model, our goal was to better understand how much of these dynamics could be explained simply by the benefit of these genes (which could also be seen as the average benefit of a gene with alternating selection pressures). We now refer back to this earlier work, to illustrate how our simple model shines light on dynamics that were first observed in a more complex eco-evolutionary context.

Whether or not the gene is beneficial may also change the mechanism of transfer, where genes that are beneficial in the long term are more likely to integrate on the chromosome and reducing the rate of loss possibly by orders of magnitude. This last point is crucial, because if the rate of loss is low enough then slightly beneficial genes could be maintained in the absence of HGT. In other words, the authors may want to consider a case where the rate of loss differs depending on the costs and benefits of the gene in a given context.

This comment is most relevant for conjugation. We now explicitly consider our results in light of transformation, and gene deletion then occurs through DNA deletion or substitution on the chromosome. While the dynamics of genes residing on plasmids vs the chromosome are indeed intriguing, it does not play a role in our current model.

This is not entirely accurate, as for any rate of gene loss > 0, there still is a class of genes that cannot be maintained without it, and can be maintained with HGT. As mentioned above, the absolute value of *l* is not particularly important, but the relationship between l and c is important. The reason why *l* and *c* are as they are in the IBM, is mainly to increase the speed of the relevant dynamics (for example, the local nucleation events from Figure 5 become very rare when scaling these rates down too much, even though they do always, eventually, occur).

A related point is how deleterious can SGEs become without disrupting HGT? This seems to be explored in Appendix 1—figure 5B and C. Is the result that SGEs would simply disappear if they are too deleterious? There seems to be a suggestion in the supplement that grid size may change this result. It would also be good to expand on this in the main text.

We have clarified in the text what happens with SGEs that are even more costly (subsection “HGT is evolutionarily maintained in the presence of harmful SGEs”). If HGT is not evolvable, nothing interesting happens when considering very costly SGEs, as the condition h > l – b still holds. Thus, no matter how costly, there is always a HGT-rate where the SGE can persist (barring SGEs with fitness penalty greater than 1.0, the intrinsic growth rate).

When HGT is evolvable (*i.e.* when we allow the individuals in the IBM to evolve their rates DNA uptake), they will decrease their rates of HGT in the presence of costly SGEs. In the spatially structured populations, this can result in interesting oscillations in the HGT-rate, as described in the main text. However, as can be seen from Figure 6, the costly SGEs go through major bottlenecks. When even costlier SGEs are considered (see Appendix 1—figure 4), the selection pressures for decreasing HGT-rates eventually purges all SGEs from the system. For β=0.04 (e.g. see the 5th column of Figure 10), the rate of HGT will continue to oscillate due to a tug-of-war between infected HGT^+^ strains and uninfected HGT^-^ strain. As discussed in the main text, the HGT^+^ strain has the advantage of being able to maintain the rescuable gene, while the HGT^-^ strain has the advantage of not being susceptible by SGEs. Due to the finite population size, the SGE is eventually lost, and the population will bounce back to higher rates of HGT (again, see column for β = 0.04). Even costlier SGEs (β = 0.05) do not persist at all, and quickly go extinct. Indeed, as mentioned, grid size does change these dynamics for β = 0.04, but we found that β = 0.05 (and greater values) could not persist even in larger populations (400x400 cells). Due to computational limitations, we have not tested what happens with even larger populations.

In sum, in order for the manuscript to be accepted in eLife, we ask the authors to at least do the following:1) Include a detailed discussion at the beginning of the paper about the assumptions of the model and which biological situation is represented. Are you simulating HGT by conjugation or transformation? Who decides whether transfer is possible – the donor, the receiver, or both?

Since one of our main research questions is “When is taking up a gene beneficial for the bacteria?”, the model best represents bacterial transformation, where cells take up DNA from their environment. We now make it explicit that we consider transformation (Introduction paragraph three, Results paragraph one and Materials and methods first paragraph), where the receiver decides whether transfer is possible. We also address the costs that this mechanism may entail (Introduction paragraph two and three).

2) The authors should more explicitly explain where their parameter choices for costs, benefits, rates of loss, etc. and how they relate to the chosen mechanism of HGT. Also, why is it reasonable to assume that they are independent of each other?

This is much clearer in the revised manuscript. For transformation, the independence of these parameters is a more reasonable assumption than for conjugation. Nevertheless, one may argue that higher rates of HGT may cause more gene loss due to illegitimate recombination / insertion into coding regions. Either way, as we do not know the precise values of these parameters and how they may correlate, the model allows us to consider the impact of 1 variable at a time.

3) Revise the choices of cost, benefit and rates of loss according to the biological situation and taking into account the ideas discussed above. Will the conclusions hold and will they be biologically relevant (if the parameter range is very small, is HGT likely to evolve)?

The statement that the parameter range is more narrow for lower loss (in the original manuscript) may have been misleading, and has been changed to avoid this confusion. Whatever the absolute loss rate, it is the relationship between loss and the costs that enable/hinders the evolution of HGT. This is now explained in the Materials and methods (section “Parameters used”). That said, *if* the parameter range for rescuable/enrichable genes is more narrow in reality, then the ability to evolve HGT will depend on the efficiency of selection (population size and mutation rates), and as mentioned by the reviewers, on the rate at which the gene for HGT is itself lost. We (and others) should take that final point into account for future modelling.

4) More fully cite the relevant literature on HGT costs and how they arise (TREE 2013 and San Millan and MacLean, Microbiology Spectrum 2017)

We now cited both these papers. As San Millan and MacLean references specifically the costs of plasmids, this is cited when comparing our results with what is known about conjugation and plasmids (i.e. Bergstrom, 2000) in the Discussion.

5) Flesh out what happens if SGEs are more deleterious.

We added descriptions of what happens with very costly SGEs in the main text. (subsection “HGT is evolutionarily maintained in the presence of harmful SGEs”)